# Stray Intrusive Outliers-Based Feature Selection on Intra-Class Asymmetric Instance Distribution or Multiple High-Density Clusters

**Lixin Yuan** [1][2]  **Yirui Wu** [1]  **Wenxiao Zhang** [1]  **Minglei Yuan** [3]  **Jun Liu** [4]

## Abstract

For data with intra-class Asymmetric instance Distribution or Multiple High-density Clusters (ADMHC), outliers are real and have specific patterns for data classification, where the class body is necessary and difficult to identify. Previous Feature Selection (FS) methods score features based on all training instances or rarely target intra-class ADMHC. In this paper, we propose a supervised FS method, Stray Intrusive Outliers-based FS (SIOFS), for data classification with intra-class ADMHC. By focusing on Stray Intrusive Outliers (SIOs), SIOFS modifies the skewness coefficient and fuses the threshold in the $3\sigma$ principle to identify the class body, scoring features based on the intrusion degree of SIOs. In addition, the refined density-mean center is proposed to represent the general characteristics of the class body reasonably. Mathematical formulations, proofs, and logical exposition ensure the rationality and universality of the settings in the proposed SIOFS method. Extensive experiments on 16 diverse benchmark datasets demonstrate the superiority of SIOFS over 12 state-of-the-art FS methods in terms of classification accuracy, normalized mutual information, and confusion matrix. SIOFS source codes is available at https://github.com/XXXly/2025-ICML-SIOFS.

## 1. Introduction

With the rapid advancement of data acquisition technologies, an increasing amount of high-dimensional data is being

[1]College of Computer Science and Software Engineering, Hohai University, Jiangsu, China [2]State Key Lab. for Novel Software Technology, Nanjing University, P.R. China [3]Hefei Institute for Advanced Research, Anhui University of Finance and Economics, Anhui, China [4]School of Computing and Communications, Lancaster University, Lancaster, United Kingdom. Correspondence to: Yirui Wu <wuyirui@hhu.edu.cn>.

*Proceedings of the $42^{nd}$ International Conference on Machine Learning*, Vancouver, Canada. PMLR 267, 2025. Copyright 2025 by the author(s).

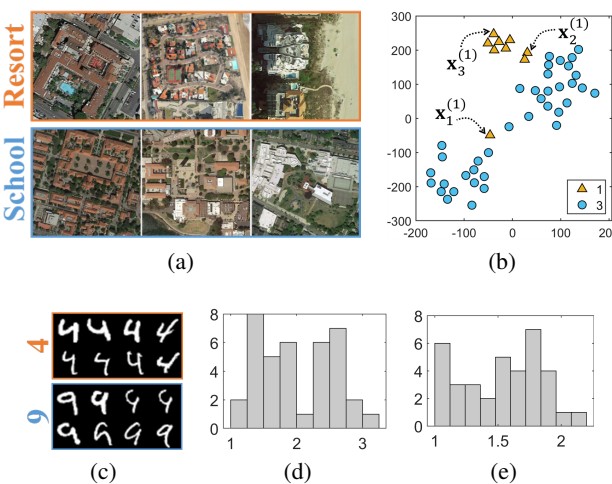

*Figure 1.* (a) Similar scene images of "Resort" and "School" on AID. (b) 2D visualization of classes "1" and "3" from CLL via T-SNE (2012). (c) Similar digits "4" and "9" on GISETTE. (d) Distance histogram of instances to center for class "3" on CLL. (e) Distance histogram of instances to center for class "1" on TOX.

generated (Li et al., 2024; Wang et al., 2024). Feature Selection (FS), the selection of a discriminative feature subset, is an essential part of data processing (Cohen et al., 2023). FS makes the representations of instances interpretable and enables fast and compact models to be learned.

High-dimensional data classification with intra-class Asymmetric instance Distribution or Multiple High-density Clusters (ADMHC) is challenging in machine learning, especially for small-sized datasets. In such tasks, misclassified outliers inevitably exist. Moreover, some outliers *intrude* into other class bodies. As shown in Fig. 1a, some instances of "Resort" share very similar structural and textural characteristics with instances of "School". Similarly, Fig. 1c shows that some instances of "4" and "9" are very similar. In Fig. 1b, $\mathbf{x}_1^{(1)}, \mathbf{x}_2^{(1)}$ are three different types of outliers from class "1", where $\mathbf{x}_1^{(1)}$ intrudes into the body of class "3", and $\mathbf{x}_2^{(1)}$ is near the body of class "3". Compared to $\mathbf{x}_2^{(1)}$, the class label of $\mathbf{x}_1^{(1)}$ is easily mispredicted as "3" by the predictive model. Focusing on $\mathbf{x}_1^{(1)}$ can reasonably iden-

tify low-ranked features for classification tasks. We define such instances like $\mathbf{x}_1^{(1)}$ as Stray Intrusive Outliers (SIOs), which are far away from the class body they belong to and intrude into other class bodies. Most current FS methods score features based on the characteristics of all training instances (Roffo et al., 2021).

Another important characteristic of high-dimensional data classification is intra-class multiple high-density clusters. As shown in Fig. 1b, class "3" has two high-density clusters. In such a class, necessarily, multiple distinct peaks exist in the distance distribution of instances to the class center (see Fig. 1d and 1e), making it difficult to determine the class center and body. Existing FS methods rarely aim to identify the class body in the context of intra-class multiple high-density clusters.

With few outliers, the mean center can reflect the overall characteristics of the class body well (Lim & Kim, 2021). Clustering by Fast search and find of Density Peaks (CFDP) (Rodriguez & Laio, 2014) acquires a density-based clustering center. However, the CFDP center is sensitive to local high density formed by a small number of instances in a class. The threshold in the $3\sigma$ principle of the normal distribution can identify the class body, but it requires the instance distribution to be centrally symmetric. The Skewness Coefficient (SC) measures the degree of asymmetry in data distribution (Vapnik, 1995), but it cannot be directly used to identify the class body. Intuitively, two classes intersect if an instance of one class intrudes into the body of another class. A general conclusion is that the sum of the radii of two hyperspheres is greater than the distance between the class centers when they intersect, which motivates us to quantify the intrusion degree of SIOs.

In this paper, our main contributions are as follows. (i) We propose a Stray Intrusive Outliers-based Feature Selection (SIOFS) method for the high-dimensional data classification with ADMHC. (ii) We present a new definition of instance density. A Refined Density-Mean (RDM) center is proposed to characterize the class body of different types of instance distributions. For the distance of intra-class instances to the center, the modified SC is fused into the threshold in the $3\sigma$ principle, reasonably identifying the class body. The formulations about the intrusion degree of SIOs are provided. Mathematical proofs or logical expositions of the settings in SIOFS are complete. (iii) Extensive comparisons on 16 various benchmark datasets show that, for selecting discriminative features, SIOFS outperforms 12 state-of-the-art methods with higher classification accuracies, normalized mutual information, and confusion matrices.

Compared with previous FS methods, SIOFS targets intra-class ADMHC for data classification, focuses on the SIOs, and scores features based on the intrusion degree of SIOs caused by features. Theorems and experiments ensure the rationality and universality of the SIOFS method.

## 1.1. Brief Related Work

FS methods select a subset of features from high-dimensional data to improve data compactness and reduce noisy features, in particular, to alleviate the overfitting, high computational cost, and low-performance issues (Nie et al., 2022; Cohen et al., 2023). There are three main FS methods: filter, wrapper and embedded (Tang et al., 2014). Filter methods evaluate each feature according to the intrinsic characteristics of the data. Wrapper and embedded methods are prone to overfitting because their selection is part of training. Most FS methods treat each instance equally during selection, rarely focusing on the outliers that lead to misclassification, and score features according to the degree of this misclassification. The FSDOC (Yuan et al., 2022) method focuses on instances within one class that are outliers in the direction of another class. The IOFS (Yuan et al., 2024) method further explores the outliers close to the other class. But importantly, both FSDOC and IOFS fail to address the FS problem in the context of ADMHC. Additional related work is shown in Appendix A.

## 2. Refined Density-Mean (RDM) Center

**Notations.** Throughout this paper, the boldface capital letters (e.g., $\mathbf{X}$) denote matrices, the boldface lowercase letters denote vectors (e.g., $\mathbf{x}$), and the italic letters are scalars (e.g., $X, x, \alpha$). $n, d, c$ denote the total numbers of training instances, feature dimensions, and classes in a dataset, respectively. Given a dataset, for class $k = 1, 2, \ldots, c$, the instance set is $\mathcal{X}^{(k)} = \{\mathbf{x}_1^{(k)}, \mathbf{x}_2^{(k)}, \ldots, \mathbf{x}_{n_k}^{(k)}\}$, where $\mathbf{x}_i^{(k)} = (x_{i1}^{(k)}, x_{i2}^{(k)}, \ldots, x_{id}^{(k)})^{\mathrm{T}} \in \mathbb{R}^d$ denotes the $i$th instance, $i = 1, 2, \ldots, n_k$, and $n_k$ is the number of training instances. To assess each feature equally, we use the $\ell_1$ norm-based distance (Yuan et al., 2022). For example, the distance between two instances $\mathbf{x}_i^{(k)}$ and $\mathbf{x}_j^{(k)}$ ($j = 1, 2, \ldots, n_k$) can be obtained by

$$\|\mathbf{x}_i^{(k)} - \mathbf{x}_j^{(k)}\|_1 = \sum\nolimits_{f=1}^{d} |x_{if}^{(k)} - x_{jf}^{(k)}|. \qquad (1)$$

**RDM Center.** Outliers in a class are defined relative to the class body, and thus, an appropriate class center to describe the general characteristics of the class body is crucial. Commonly, the class body is characterized by high-density instances and contains more than half of the total instances. Considering an imaginary hypersphere centered at instance $\mathbf{x}_i^{(k)}$, which contains at least half of the instances in class $k$, we define the *instance density* of $\mathbf{x}_i^{(k)}$ as the reciprocal of the radius of this hypersphere. Combining the advantages of density-based and mean centers, we average the instances with relatively high density as the proposed RDM center.

The details of obtaining the RDM center are as follows.

First, for class $k$ and $n_k \geq 3$, we compute the $\ell_1$-norm distance $d_{ij}^{(k)} = \|\mathbf{x}_i^{(k)} - \mathbf{x}_j^{(k)}\|_1$ between $\mathbf{x}_i^{(k)}$ and $\mathbf{x}_j^{(k)}$ ($i, j = 1, 2, \ldots, n_k$). The $i$th row of the matrix $(d_{ij}^{(k)})_{n_k \times n_k}$ is the distance from $\mathbf{x}_i^{(k)}$ to each instance of class $k$.

Second, for $i = 1, 2, \ldots, n_k$, let $\varepsilon_i^{(k)} = \text{median}(d_{i1}^{(k)}, d_{i2}^{(k)}, \ldots, d_{in_k}^{(k)})$, where $\text{median}(\mathcal{X})$ acquires the median of elements in $\mathcal{X}$. Consequently, the hypersphere centered at $\mathbf{x}_i^{(k)}$ with radius $\varepsilon_i^{(k)}$ contains no less than $\frac{n_k}{2}$ instances (including $\mathbf{x}_i^{(k)}$ itself) of class $k$. The smaller the value of $\varepsilon_i^{(k)}$, the higher the density of $\mathbf{x}_i^{(k)}$.

Third, due to the fact that the ratio of higher density to all instances in a class is obviously different for different classes, we introduce a ratio $\alpha \in (0, 1]$ to construct the higher-density instance threshold $T^{(k)}$ as

$$T^{(k)} = \max(\text{mint}(\{\varepsilon_1^{(k)}, \varepsilon_2^{(k)}, \ldots, \varepsilon_{n_k}^{(k)}\}, \lceil \alpha \cdot n_k \rceil)). \quad (2)$$

In (2), $\text{mint}(\mathcal{S}, t)$ finds the $t$ smallest elements of set $\mathcal{S}$, and $\lceil \cdot \rceil$ is a round up operation. Let $\mathcal{H}^{(k)}$ denote the subset of higher density instances of class $k$. The instance $\mathbf{x}_i^{(k)}$ can be selected into $\mathcal{H}^{(k)}$ if it obeys $\varepsilon_i^{(k)} \leq T^{(k)}$.

Finally, the RDM center $\mathbf{u}^{(k)} \in \mathbb{R}^d$ of class $k$ is

$$\mathbf{u}^{(k)} = \frac{1}{|\mathcal{H}^{(k)}|} \sum_{\mathbf{x}_i^{(k)} \in \mathcal{H}^{(k)}} \mathbf{x}_i^{(k)}, \quad (3)$$

where $|\mathcal{H}^{(k)}|$ is the size of $\mathcal{H}^{(k)}$. In (3), the mean operation appropriately eliminates random errors and captures the general characteristics of the class body.

Based on the sparsity of outliers and the definition of instance density, outliers have low instance density and are not allowed to enter $\mathcal{H}^{(k)}$ unless $\alpha$ is large in $(0, 1]$. Two special cases are as follows: The RDM center is (i) the instance itself when $n_k = 1$ and (ii) the average of two instances when $n_k = 2$. In particular, the RDM center is dimension-independent and can be calculated for a series of 1-dimensional numbers. We formulate the procedure for obtaining the RDM center as $\text{RDM}(\mathcal{X}, \alpha)$, where $\mathcal{X}$ represents the input dataset, and $\alpha$ is derived from (2).

## 3. Pattern of Stray Intrusive Outliers and Feature Selection Method

### 3.1. Identifying Stray Intrusive Outliers

As shown in Fig. 1b, the SIO intuitively intrudes other class body and intersects the instances in other class.

**Recognizing potential SIO of class $k$ towards class $l$:** For class $l = 1, 2, \ldots, c$, we calculate the RDM center $\mathbf{u}^{(l)} =$

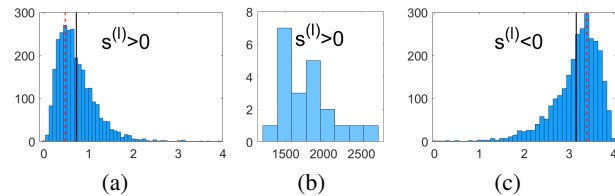

*Figure 2.* Histograms of distances between instances to the center. The average is denoted by the solid black line and the mode is denoted by the red dashed line. (a) Data from Class "2" of GISETTE. (b) Data from Class "3" of Carcinom. (c) Synthetic data.

$\text{RDM}(\mathcal{X}^{(l)}, \alpha)$ and the distance between $\mathbf{x}_i^{(l)}$ and $\mathbf{u}^{(l)}$ via $d_i^{(l)} = \|\mathbf{x}_i^{(l)} - \mathbf{u}^{(l)}\|_1, i = 1, 2, \ldots, n_l$.

To mine the intrusive relationship from class $k$ to class $l$, it is necessary to identify the body of class $l$. Essentially, a segmentation threshold $\Theta^{(l)}$ is required for $d_1^{(l)}, d_2^{(l)}, \ldots, d_{n_l}^{(l)}$, satisfying the conditions that (i) the body of class $l$ can be extracted, and (ii) when $\mathbf{x}_i^{(k)} \in \mathcal{X}^{(k)}$ meets $\|\mathbf{x}_i^{(k)} - \mathbf{u}^{(l)}\|_1 < \Theta^{(l)}$, $\mathbf{x}_i^{(k)}$ intersects the instances in class $l$. Condition (i) is equivalent to more than half of $d_1^{(l)}, d_2^{(l)}, \ldots, d_{n_l}^{(l)}$ being less than $\Theta^{(l)}$, and thus $\Theta^{(l)}$ should be larger. However, as shown in Fig. 1b, if $\Theta^{(3)}$ is large enough to make $\|\mathbf{x}_3^{(1)} - \mathbf{u}^{(3)}\|_1 < \Theta^{(3)}$, but $\mathbf{x}_3^{(1)}$ does not intrude into the body of class "3". That is, condition (ii) requires a smaller $\Theta^{(l)}$.

The $3\sigma$ principle of normal variable that the probability $Pr(\xi > \mu + 3\sigma) < 0.002$ when $\xi \sim N(\mu, \sigma^2)$ (Vapnik, 1995) provides a probability-based threshold $\mu + 3\sigma$. However, this threshold is more suitable when the distribution of $d_1^{(l)}, d_2^{(l)}, \ldots, d_{n_l}^{(l)}$ is symmetric. Additionally, the larger coefficient "3" causes $\mu + 3\sigma$ to fail in satisfying the condition (ii) of $\Theta^{(l)}$. In high-dimensional data classification with intra-class ADMHC, consequentially, the distribution of $d_1^{(l)}, d_2^{(l)}, \ldots, d_{n_l}^{(l)}$ is asymmetry or multi-peak.

The common *Skewness Coefficient* (SC) embodies the asymmetry degree of data w.r.t its mean, and the SC of $d_1^{(l)}, d_2^{(l)}, \ldots, d_{n_l}^{(l)}$ (denoted as $s^{(l)}$), usually $s^{(l)} \in (-3, 3)$ (Linton, 2017). The larger the value of $|s^{(l)}|$, the greater the asymmetry degree of the data distribution. Based on the statistical theory, we have the conclusion that "*mode*[1] < *average*" if $s^{(l)} > 0$, and "*average* < *mode*" if $s^{(l)} < 0$ (the prototype and properties of SC are shown in Appendix B).

For the representative ability of the body of $d_1^{(l)}, d_2^{(l)}, \ldots, d_{n_l}^{(l)}$, the RDM center is superior to the mean. Let $\text{u}^{(l)} = \text{RDM}(\{d_1^{(l)}, d_2^{(l)}, \ldots, d_{n_l}^{(l)}\}, \alpha)$, the corresponding standard

---

[1]Mode is the value of the highest frequency in the statistical distribution and also the highest central tendency.

deviation $\hat{\sigma}^{(l)} = \sqrt{\frac{1}{n_l} \sum_{i=1}^{n_l} (d_i^{(l)} - \mathrm{u}^{(l)})^2}$. To avoid many parameters, we have the same $\alpha$ value for acquiring $\mathrm{u}^{(l)}$ and the class center $\mathbf{u}^{(l)}$, because both $\alpha$ values reflect the dispersion degree of instances within the same class $l$. Thus, for $d_1^{(l)}, d_2^{(l)}, \ldots, d_{n_l}^{(l)}$, the modified SC is formulated as

$$\hat{s}^{(l)} = \frac{1}{n_l (\hat{\sigma}^{(l)})^3} \sum_{i=1}^{n_l} (d_i^{(l)} - \mathrm{u}^{(l)})^3. \qquad (4)$$

Based on the procedure of obtaining the RDM center, the RDM center $\mathrm{u}^{(l)}$ of $d_1^{(l)}, d_2^{(l)}, \ldots, d_{n_l}^{(l)}$ is equal to the mean when $\alpha = 1$, namely that $s^{(l)}$ is a special case of $\hat{s}^{(l)}$. Thus, $\hat{s}^{(l)}$ has the property of $s^{(l)}$. The mathematical supports about the final $\Theta^{(l)}$ are in Theorem 1 and 2. The proofs of Theorem 1 and 2 are deferred to Appendix C.

**Theorem 1.** *Let $\xi$ be the distance between the instance and the center in class $l$, $\xi$ is a continuous random variable. $d_1^{(l)}, d_2^{(l)}, \ldots, d_{n_l}^{(l)}$ is a random sample of $\xi$. The meanings of $\mathrm{u}^{(l)}$ and $\hat{\sigma}^{(l)}$ are given in (4). When $\hat{\sigma}^{(l)} > 0$, we have the probability $Pr(\xi < \mathrm{u}^{(l)} + 2\hat{\sigma}^{(l)}) > \frac{3}{4}$.*

**Theorem 2.** *For $d_1^{(l)}, d_2^{(l)}, \ldots, d_{n_l}^{(l)}$, $\mathrm{u}^{(l)}, \hat{s}^{(l)}$ are the same as in (4), and $mode^{(l)}$ is the same as the footnote of Section 3.1. When $\alpha \in (0, 1]$, $mode^{(l)} \leq \mathrm{u}^{(l)} \leq average^{(l)}$ holds with probability 1 if $\hat{s}^{(l)} > 0$, and $average^{(l)} \leq \mathrm{u}^{(l)} \leq mode^{(l)}$ holds with probability 1 if $\hat{s}^{(l)} < 0$.*

In Theorem 1, $\hat{\sigma}^{(l)} > 0$ holds when class $l$ contains at least two different instances. Based on Theorem 1, $\mathrm{u}^{(l)} + 2\hat{\sigma}^{(l)}$ is greater than most of $d_1^{(l)}, d_2^{(l)}, \ldots, d_{n_l}^{(l)}$. However, it is improper to ignore different asymmetric distributions under $\hat{s}^{(l)} > 0$ or $\hat{s}^{(l)} < 0$ and employ the same coefficient "2". Based on Theorem 2, $\mathrm{u}^{(l)} + 2\hat{\sigma}^{(l)}$ is relatively large for obtaining the highest density value in $d_1^{(l)}, d_2^{(l)}, \ldots, d_{n_l}^{(l)}$ when $\hat{s}^{(l)} > 0$. Similarly, if $\hat{s}^{(l)} < 0$, $\mathrm{u}^{(l)} + 2\hat{\sigma}^{(l)}$ is relatively small for obtaining the highest density value in $d_1^{(l)}, d_2^{(l)}, \ldots, d_{n_l}^{(l)}$ when $\hat{\sigma}^{(l)} < \frac{1}{2}(mode^{(l)} - \mathrm{u}^{(l)})$.

The two contradictory conditions of $\Theta^{(l)}$ demand that, we should employ the high density values in $d_1^{(l)}, d_2^{(l)}, \ldots, d_{n_l}^{(l)}$. Consequently, we need to reduce $\mathrm{u}^{(l)} + 2\hat{\sigma}^{(l)}$ if $\hat{s}^{(l)} > 0$ and increase $\mathrm{u}^{(l)} + 2\hat{\sigma}^{(l)}$ if $\hat{s}^{(l)} < 0$. Usually, $s^{(l)} \in (-3, 3)$, $\frac{s^{(l)}}{3} \in (-1, 1)$, and $s^{(l)}$ is a special case of $\hat{s}^{(l)}$. For simplicity, whether $\hat{s}^{(l)} > 0$ or $\hat{s}^{(l)} < 0$, we normalize $\hat{s}^{(l)}$ with the constant "3", i.e., $\frac{\hat{s}^{(l)}}{3}$, to slightly regulate the coefficient "2" and assign the $2 - \frac{1}{3}\hat{s}^{(l)}$ to $\hat{\sigma}^{(l)}$. That is, we uniformly formulate $\Theta^{(l)}$ as

$$\Theta^{(l)} = \mathrm{u}^{(l)} + (2 - \frac{\hat{s}^{(l)}}{3})\hat{\sigma}^{(l)}. \qquad (5)$$

*More importantly*, when the distribution of $d_1^{(l)}, d_2^{(l)}, \ldots, d_{n_l}^{(l)}$ is multi-peak, it is particularly valuable to replace the coefficient "2" with $2 - \frac{1}{3}\hat{s}^{(l)}$. The highest peak is on the left when $\hat{s}^{(l)} > 0$, while on the right when $\hat{s}^{(l)} < 0$ (see Fig. 1d

under $\hat{s}^{(l)} = 1.37 > 0$ and Fig. 1e under $\hat{s}^{(l)} = -1.30 < 0$). In these cases, $\mathrm{u}^{(l)} + (2 - \frac{\hat{s}^{(l)}}{3})\hat{\sigma}^{(l)}$ is more appropriate than $\mathrm{u}^{(l)} + 2\hat{\sigma}^{(l)}$ on capturing the values around the highest peak whether $\hat{s}^{(l)} > 0$ or $\hat{s}^{(l)} < 0$.

Here, three statistical indexes (i.e., $\mathrm{u}^{(l)}, \hat{\sigma}^{(l)}, \hat{s}^{(l)}$) that we introduce into (5) can reflect more statistical properties of data and ensure the general applicability to different types of instance distributions.

About recognizing potential SIO of class $k$ towards class $l$, for $\mathbf{x}_i^{(k)} \in \mathcal{X}^{(k)}$, if there exists class $l$ ($l \neq k$) that meets

$$\|\mathbf{x}_i^{(k)} - \mathbf{u}^{(l)}\|_1 < \mathrm{u}^{(l)} + (2 - \frac{\hat{s}^{(l)}}{3})\hat{\sigma}^{(l)}, \qquad (6)$$

then $\mathbf{x}_i^{(k)}$ is identified as the potential SIO of class $k$ towards class $l$. Note that if more than one $l$ (denoted as $l_1, l_2, \ldots$) satisfies (6), intuitively, $\mathbf{x}_i^{(k)}$ is easily misclassified as the class whose center is closest to $\mathbf{x}_i^{(k)}$. Therefore, we identify $\mathbf{x}_i^{(k)}$ as the *potential SIO* of class $k$ towards the unique class $l_0 = \arg \min_{l \in \{l_1, l_2, \ldots\}} (\|\mathbf{x}_i^{(k)} - \mathbf{u}^{(l)}\|_1)$, and all these potential SIOs form the set $\mathcal{X}^{(kl_0)}$. In particular, (6) is independent of the data structures of classes $k$ and $l_0$.

**Examining whether $\mathbf{x}_i^{(k)} \in \mathcal{X}^{(kl_0)}$ is the SIO from class $k$ towards class $l_0$ if $\mathcal{X}^{(kl_0)} \neq \emptyset$:** When instances intrude from one class into another, it indicates that the two classes intersect. Considering the non-spherical data structures of the classes, we follow the general conclusion in Sec. 1 with the oriented radius from one class to another.

Let $\mathcal{X}^{(l_0)k} = \{\mathbf{x}_i^{(l_0)} : \|\mathbf{x}_i^{(l_0)} - \mathbf{u}^{(k)}\|_1 < \Theta^{(k)}, i = 1, 2, \ldots, n_{l_0}\}$ denote the instance subset where $\mathbf{x}_i^{(l_0)} \in \mathcal{X}^{(l_0)}$ but enters the body of class $k$. Note that, for same $l_0$ but different $k$, $\mathcal{X}^{(l_0)k}$ may contain the same instances. This makes $\mathcal{X}^{(l_0)k}$ different from $\mathcal{X}^{(l_0 k)}$. Thus, if $\mathcal{X}^{(l_0)k} \neq \emptyset$, the radius $D^{(l_0)k}$ of class $l_0$ towards class $k$ is calculated by

$$D^{(l_0)k} = \frac{1}{|\mathcal{X}^{(l_0)k}|} \sum_{\mathbf{x}_i^{(l_0)} \in \mathcal{X}^{(l_0)k}} \|\mathbf{x}_i^{(l_0)} - \mathbf{u}^{(l_0)}\|_1. \qquad (7)$$

The averaging operation in (7) can also properly eliminate the random errors in the data. If $\mathbf{x}_i^{(k)} \in \mathcal{X}^{(kl_0)}$ meets

$$\|\mathbf{x}_i^{(k)} - \mathbf{u}^{(k)}\|_1 + D^{(l_0)k} - \|\mathbf{u}^{(k)} - \mathbf{u}^{(l_0)}\|_1 > 0, \qquad (8)$$

we identify $\mathbf{x}_i^{(k)}$ as the *SIO* of class $k$ towards class $l_0$ ($k \neq l_0$). All these SIOs form the SIOs set $\mathcal{X}_o^{(kl_0)}$. If $\mathcal{X}_o^{(kl_0)} \neq \emptyset$, the classes $k, l_0$ are recognized as a *SIO class pair*. In addition, there is no intrusive relation between class $k$ and $l_0$ if either $\mathcal{X}^{(l_0)k} = \emptyset$ or $\mathcal{X}_o^{(kl_0)} = \emptyset$.

## 3.2. Feature Selection on Stray Intrusive Outliers

We further assess SIOs in $\mathcal{X}_o^{(kl_0)}$ and the SIO class pair $k, l_0$ on feature level. Let $\mathbf{x}_i^{(k)} = (x_{i1}^{(k)}, x_{i2}^{(k)}, \ldots, x_{id}^{(k)})^{\mathrm{T}} \in \mathcal{X}_o^{(kl_0)}$

and class center $\mathbf{u}^{(k)} = (u_1^{(k)}, u_2^{(k)}, \ldots, u_d^{(k)})^{\mathrm{T}}$. Considering $\sum_i \sum_j a_{ij} = \sum_j \sum_i a_{ij}$ and (1), (7) (8) are rewritten as

$$D^{(l_0)k} = \sum_{f=1}^d \frac{1}{|\mathcal{X}^{(l_0)k}|} \sum_{\mathbf{x}_i^{(l_0)} \in \mathcal{X}^{(l_0)k}} |x_{if}^{(l_0)} - u_f^{(l_0)}|, \quad (9)$$

$$\sum_{f=1}^d (|x_{if}^{(k)} - u_f^{(k)}| + D_f^{(l_0)k} - |u_f^{(k)} - u_f^{(l_0)}|) > 0. \quad (10)$$

We denote $D_f^{(l_0)k} = \frac{1}{|\mathcal{X}^{(l_0)k}|} \sum_{\mathbf{x}_i^{(l_0)} \in \mathcal{X}^{(l_0)k}} |x_{if}^{(l_0)} - u_f^{(l_0)}|$ and $S_{if}^{(kl_0)} = |x_{if}^{(k)} - u_f^{(k)}| + D_f^{(l_0)k} - |u_f^{(k)} - u_f^{(l_0)}|$. They are the $f$th terms of (9) and (10) from feature $f$, respectively.

At the level of feature $f = 1, 2, \ldots, d$, $|x_{if}^{(k)} - u_f^{(k)}|$ is the distance between $\mathbf{x}_i^{(k)}$ and $\mathbf{u}^{(k)}$, $D_f^{(l_0)k}$ is the average distance from $\mathbf{x}_i^{(l_0)} \in \mathcal{X}^{(l_0)k}$ to $\mathbf{u}^{(l_0)}$, where $\mathbf{x}_i^{(l_0)}$ is in class $l_0$ but enter the body of class $k$. $|u_f^{(k)} - u_f^{(l_0)}|$ is the distance between the two class centers. Inspired by the expectation of small intra-class diversity and large inter-class scatter in classification tasks (Xu et al., 2022), $|x_{if}^{(k)} - u_f^{(k)}|$ and $D_f^{(l_0)k}$ should be small, and $|u_f^{(k)} - u_f^{(l_0)}|$ should be large if feature $f$ is sufficiently discriminative. Consequently, for each $\mathbf{x}_i^{(k)} \in \mathcal{X}_o^{(kl_0)}$, we expect $S_{if}^{(kl_0)} < 0$ with $|S_{if}^{(kl_0)}|$ being large. Thus, a reasonable FS strategy is to assign a higher rank to feature $f$ whose $S_{if}^{(kl_0)}$ appears at the top of the ascending order.

**FS on one SIO class pair $k, l_0$:** To reflect the overall performance of the feature $f$ on all SIOs in $\mathcal{X}_o^{(kl_0)}$, we average all $S_{if}^{(kl_0)}$ values as $\bar{S}_f^{(kl_0)} = \frac{1}{|\mathcal{X}_o^{(kl_0)}|} \sum_{\mathbf{x}_i^{(k)} \in \mathcal{X}_o^{(kl_0)}} S_{if}^{(kl_0)}$. On the other hand, for the SIO class pair $k, l_0$, $\sum_{f=1}^d \bar{S}_f^{(kl_0)}$ and $\bar{S}_f^{(kl_0)}$ reflect the average intrusion degree on all features and one feature, respectively. The larger the values of $\bar{S}_1^{(kl_0)}, \bar{S}_2^{(kl_0)}, \ldots, \bar{S}_d^{(kl_0)}$, the greater their contribution to $\sum_{f=1}^d \bar{S}_f^{(kl_0)}$. Therefore, the appropriate evaluation criterion should rank the features from highest to lowest by sorting $\bar{S}_1^{(kl_0)}, \bar{S}_2^{(kl_0)}, \ldots, \bar{S}_d^{(kl_0)}$ in ascending order. The theoretical foundation for the rationality of this evaluation criterion is shown in Appendix D.

**FS on all SIO class pairs:** Real-world classification tasks seek to achieve higher overall classification accuracy while minimizing the intrusion degree between all class pairs. For the class pair with maximum intrusion degree, although many features can be discarded to reduce the degree, this will result in an inadequate representation of the instances from other classes. Taken together, we consider the half of all SIO class pairs with the lowest intrusion degree and score feature $f$ by averaging the $\bar{S}_f^{(kl_0)}$ values on this subset of class pairs. For a dataset, let $N_{ip}$ denote the number of all SIO class pairs identified in Sec. 3.1.

First, we compute $\sum_{f=1}^d \bar{S}_f^{(kl_0)}$ for each SIO class pair $k, l_0$. Second, the $\lceil 0.5N_{ip} \rceil$ smallest sums are selected, and

the corresponding vector $(\bar{S}_1^{(kl_0)}, \bar{S}_2^{(kl_0)}, \ldots, \bar{S}_d^{(kl_0)})$ forms a row of the matrix $\mathbf{P} = (P_{ij})_{\lceil 0.5N_{ip} \times d \rceil}$. Finally, let $s_f$ ($f = 1, 2, \ldots, d$) represent the score of the $f$th feature. We can obtain the feature scores by

$$(s_1, \ldots, s_d) = \frac{1}{\lceil 0.5N_{ip} \rceil} \left( \sum_{i=1}^{\lceil 0.5N_{ip} \rceil} P_{i1}, \ldots, \sum_{i=1}^{\lceil 0.5N_{ip} \rceil} P_{id} \right). \quad (11)$$

Note that the $f$th feature has no discriminability between the two classes if its values are the same across all instances from both classes. In this case, we assign a score of $+\infty$ to such a feature. The scores $s_1, s_2, \ldots, s_d$ are then ranked in ascending order, and the top $m$ features are selected.

**Algorithms and time complexity.** Obtaining the RDM center, the SIOs set, and the final FS cost $\mathcal{O}(n^2+n)$, $\mathcal{O}(nc+c^2)$ and $\mathcal{O}(dc)$ time, respectively. The corresponding algorithms and time complexity analysis for each step are deferred to Appendix E.

# 4. Experiments and Analyses

## 4.1. Experimental Settings

**Baselines:** We use 12 state-of-the-art supervised FS methods as baselines, along with three popular unsupervised FS methods for extensions. The names, types, and references of all FS methods are given in Appendix F. For fair comparisons, parameter settings and codes for all methods are directly taken from their original publications.

**Datasets:** Sixteen multi-type datasets are considered due to their classification challenges, including intra-class ADMHC and variations in the number of instances, features, and classes. Detailed information about the 16 datasets is given in Appendix F. For Dataset #1∼#12 in Table 6 (see Appendix F), instances are represented by low-level features. For Dataset #13∼#15 in Table 6, deep features are extracted using configurations similar to (Yuan et al., 2024). For #16 Caltech101 dataset, high-level visual feature, i.e., Fisher vector (FV), is extracted (Zhang et al., 2014; Yuan et al., 2022).

**Experimental protocol:** We use 3 widely accepted metrics: *accuracy* (ACC), *normalized mutual information* (NMI) (Cai et al., 2011), and confusion matrix (CM) to evaluate the performance of the same features selected by the FS methods on a dataset. Higher values of ACC, NMI, and the sum of the principal diagonal of CM indicate better classification performance of selected features.

For datasets represented by low-level features, the classification tasks aim to achieve better performance with fewer selected features. Thus, for each FS method, the ACC and NMI results are computed under the top $50, 100, \ldots, 300$ features, respectively. The mean and standard deviation

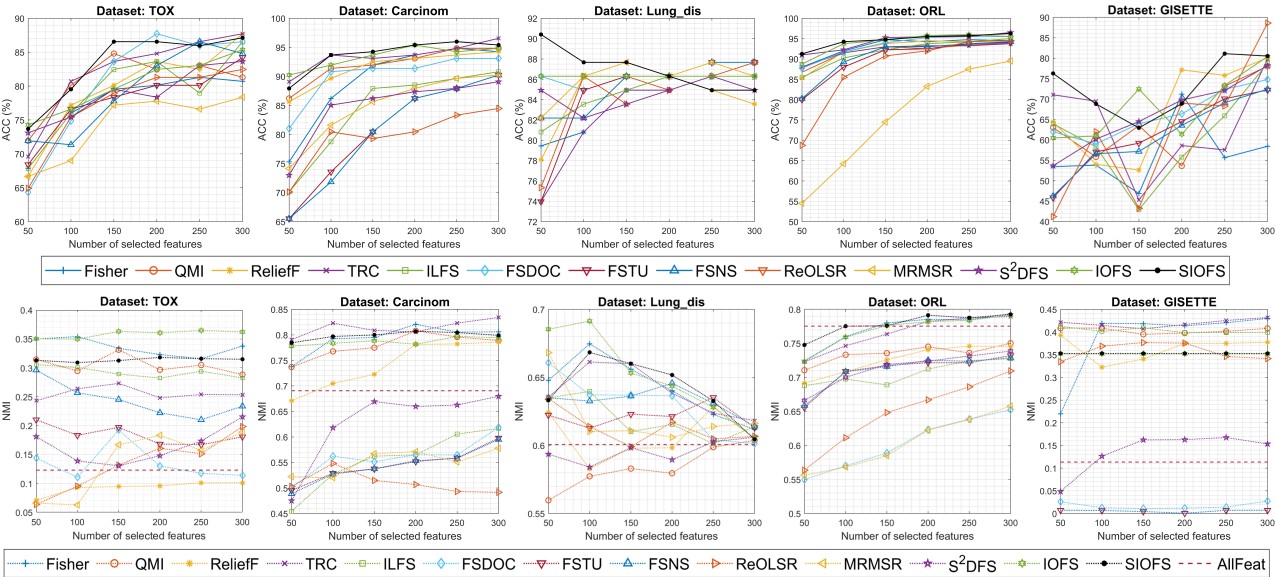

*Figure 3.* ACC and NMI results of FS methods on some datasets w.r.t the top 50, . . . , 300 features. AllFeat: All features are selected.

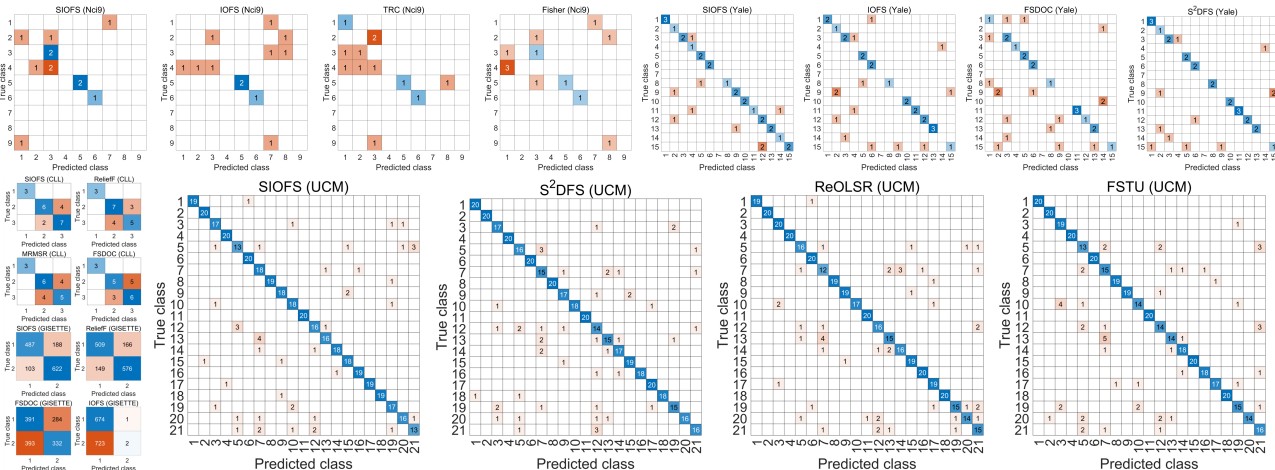

*Figure 4.* CMs on some datasets. The selected methods correspond to the largest four $\overline{acc}$ results in Table 1 for each dataset. SIOFS has the same $\alpha$ as in Table 1 for all datasets. 100 or 300 features are employed.

of six ACC results are denoted as $\overline{acc} \pm std$. For datasets represented by deep features, the classification task focuses on achieving the highest classification accuracy regardless of the number of selected features. Hence, for each FS method, both ACC and NMI results are calculated with the top $5\%, 10\%, \ldots, 95\%$ of all features, and the best global ACC and corresponding NMI are reported. When the RDM center is involved, its parameter $\alpha$ is turned over $0.05, 0.1, \ldots, 0.95$.

Similar to most related work, we use the linear SVM (Guyon et al., 2002) as the classifier for ACC and CM. To be repeatable, we adopt the same strategy as in the literature (Yuan et al., 2024) to replace the random functions in NMI algorithms with deterministic ones. All experiments employ

fivefold cross-validation, except for ModelNet, where we use its partitioned training and test sets directly. In the following tables, the best results are marked in bold, and the second-best results are underlined.

### 4.2. Results and Discussions

**Challenge 1: small-sized high-dimensional.** The first 11 datasets (see Table 6 in Appendix F) are typical small-sized high-dimensional scenarios. Due to intra-class ADMHC and the small sample size, the main difficulty in classifying such data is identifying the true characteristics of class instances. Figure 3 and 4 show the results of ACC, NMI, and CM on some datasets, while others are deferred to Appendix G, Fig. 7. The following conclusions can be drawn:

*Table 1.* Mean and standard deviation ($\overline{acc}\pm std$) of ACC (%) w.r.t the top 50, 100, . . . , 300 features on the 12 datasets. "N/A": ReOLSR does not converge on GLIOMA and MRMSR code runs very slowly on GISETTE. $\langle\alpha\rangle$ shows the parameter $\alpha$ of SIOFS.

| Baseline | CLL | TOX | Carcinom | Lung | Lung_dis | Lymphoma | Nci9 | GLIOMA | colon | ORL | Yale | GISETTE |
|---|---|---|---|---|---|---|---|---|---|---|---|---|
| Fisher | 57.66±2.99 | 77.58±4.67 | 89.37±6.92 | 93.43±1.41 | 83.79±2.67 | 83.86±3.54 | 51.67±6.01 | 72.00±6.00 | 77.69±3.01 | 93.04±2.45 | 67.98±2.41 | 56.57±7.42 |
| QMI | 64.72±1.98 | 80.60±4.14 | 92.05±2.92 | 94.66±0.77 | 85.62±1.72 | 85.59±2.44 | 47.22±5.66 | 70.33±2.92 | 79.30±2.36 | 91.83±2.39 | 68.89±4.15 | 64.53±8.71 |
| ReliefF | 69.22±4.04 | 79.63±6.63 | 91.48±2.99 | 93.76±0.37 | 84.47±3.13 | 86.46±2.33 | 51.67±6.31 | 68.67±4.27 | 81.18±6.15 | 92.33±3.31 | 63.64±9.87 | 67.35±11.06 |
| TRC | 59.31±2.40 | 82.16±6.05 | 93.49±2.27 | 94.17±0.96 | 82.88±4.39 | 86.46±2.25 | 54.17±3.30 | 73.67±0.75 | 81.72±1.78 | 92.13±1.97 | 64.65±5.40 | 63.63±11.17 |
| ILFS | 55.40±3.20 | 79.24±6.10 | 84.29±7.46 | 90.40±1.23 | 84.70±2.00 | 85.25±3.63 | 32.50±5.51 | 71.33±3.94 | 81.18±2.58 | 91.83±3.22 | 59.90±7.01 | 60.92±10.87 |
| FSDOC | 65.01±0.96 | 80.51±3.84 | 90.13±4.16 | 91.38±5.62 | 86.07±0.51 | 84.20±5.08 | 53.33±5.61 | 69.67±5.47 | 70.97±3.36 | 93.29±2.80 | 70.91±1.82 | 66.42±5.56 |
| FSTU | 61.71±2.79 | 77.97±4.85 | 80.65±8.71 | 91.63±4.73 | 84.47±4.79 | 89.24±1.67 | 36.67±6.01 | 68.33±5.82 | 82.80±4.02 | 90.08±4.92 | 63.33±7.83 | 61.47±8.83 |
| FSNS | 64.41±4.64 | 79.24±6.01 | 80.37±8.96 | 90.72±4.31 | 85.39±2.33 | 89.58±1.35 | 36.67±6.01 | 68.33±6.26 | 82.53±4.31 | 90.58±4.89 | 60.00±5.20 | 60.84±8.70 |
| ReOLSR | 61.71±3.89 | 77.39±6.03 | 79.69±4.64 | 93.76±0.84 | 84.02±4.08 | 85.76±5.33 | 39.17±6.85 | N/A | 80.38±2.54 | 87.63±8.95 | 59.09±6.80 | 62.06±16.23 |
| MRMSR | 65.32±2.93 | 74.27±4.63 | 84.87±5.59 | 90.64±0.81 | 86.07±1.84 | 82.12±7.54 | 42.22±6.78 | 75.67±1.80 | 74.46±4.31 | 75.58±12.71 | 53.64±6.35 | N/A |
| S²DFS | 63.81±4.89 | 78.85±3.78 | 84.77±5.42 | 93.19±0.34 | 84.47±1.29 | 88.20±2.14 | 41.95±7.54 | 72.33±4.07 | 76.08±4.31 | 94.33±1.99 | 70.81±4.92 | 66.40±8.04 |
| IOFS | 63.21±6.05 | 80.21±3.84 | 93.39±1.78 | 94.66±2.04 | 86.30±0.00 | 88.37±3.03 | 54.72±3.39 | 74.00±2.00 | 79.84±4.34 | 94.08±2.50 | 71.31±2.26 | 68.15±7.66 |
| SIOFS | **71.32±3.84** | **83.24±4.99** | **93.77±2.72** | **95.32±1.02** | **86.99±1.90** | **89.76±2.20** | 54.17±2.10 | **76.00±2.00** | **83.33±1.20** | **94.63±1.64** | 71.21±4.70 | **73.08±6.70** |
| $\langle\alpha\rangle$ | $\langle 0.30\rangle$ | $\langle 0.30\rangle$ | $\langle 0.55\rangle$ | $\langle 0.50\rangle$ | $\langle 0.30\rangle$ | $\langle 0.30\rangle$ | $\langle 0.10\rangle$ | $\langle 0.15\rangle$ | $\langle 0.60\rangle$ | $\langle 0.10\rangle$ | $\langle 0.55\rangle$ | $\langle 0.10\rangle$ |

(i) Across varying numbers of selected features, SIOFS almost always provides comparable or superior results to other FS methods. In particular, when the top 50 features are selected, SIOFS outperforms the second-best by 5.48% on Lung_dis. (ii) SIOFS achieves larger NMI results compared to the baselines, ranking higher on the line chart. (iii) The ACC and NMI results of SIOFS fluctuate less than these of the baselines, indicating that SIOFS is less affected by the number of selected features and data distributions. (iv) Based on the CMs in Fig. 4, SIOFS can correctly predict more instances that are challenging for the baselines, particularly for classes with notable ADMHC, such as class "3" on CLL and class "2" on GISETTE (see Fig. 1b and 2a).

To further demonstrate the superiority of SIOFS, Table 1 summarizes the corresponding $\overline{acc}\pm std$ results for each FS method. It can be seen that SIOFS achieves the best or second-best performance on all datasets.

Note that, due to the imbalanced classes in Lymphoma and Nci9 datasets, at one of the fivefold cross-validation, we have only one training instance of one class. When $n_l = 1$, for (5), $\mathbf{u}^{(l)} = \mathbf{x}_1^{(l)}$, $d_1^{(l)} = \|\mathbf{x}_1^{(l)} - \mathbf{u}^{(l)}\|_1 = 0$, thus $\mathbf{u}^{(l)} = \text{RDM}(\{d_1^{(l)}, d_2^{(l)}, \ldots, d_{n_l}^{(l)}\}, \alpha) = 0$, $\hat{\sigma}^{(l)} = 0$, and $\Theta^{(l)} = 0$. At this situation, let $\Theta^{(l_0)} = 0$ and $\Theta^{(1)}, \ldots, \Theta^{(c_0)}$ $(c_0 < c)$ be the ones larger than 0 according to (5), we employ the variable coefficient (denoted as $\nu$) of $\Theta^{(1)}, \ldots, \Theta^{(c_0)}$, and reset $\Theta^{(l_0)} = \nu(c-1) \cdot \min(\Theta^{(1)}, \ldots, \Theta^{(c_0)})$, where the variable coefficient $\nu$ of data is the ratio of the standard deviation to the mean, i.e., $\nu = \frac{\sigma}{\mu}$ and reflects the degree of data dispersion (Vapnik, 1995).

**Challenge 2: severe inter-class intrusion.** Recently, mining valuable information in the context of severe inter-class intrusion is necessary in some AI application domains (Deng et al., 2024; Kumar & Kumar, 2024). Here, we consider a handwritten digits classification dataset from the NIPS FS challenge (GISETTE). The corresponding results of ACC, NMI, and CM on GISETTE are reported in Fig. 3 and Fig. 4, and the $\overline{acc}\pm std$s are shown in Table 1. SIOFS achieves superior performance in terms of ACC, NMI, and CM. For

*Table 2.* $\overline{acc}\pm std$ of SIOFS with RDM, CFDP and MEAN centers. And the $\overline{acc}$ difference of SIOFS and w/o SC. "N/A": the SIOs can not be captured by MEAN center.

| Dataset | RDM $\alpha = 0.1$ | CFDP | MEAN | w/o SC | $\langle\alpha\rangle$ | SIOFS -w/o SC |
|---|---|---|---|---|---|---|
| CLL | **66.37±3.15** | 65.02±3.87 | 58.71±2.41 | 68.47±1.73 | $\langle 0.20\rangle$ | 2.85 |
| TOX | **79.82±2.54** | 76.51±3.37 | 74.56±2.54 | 82.07±3.20 | $\langle 0.15\rangle$ | 1.17 |
| Carcinom | 91.48±1.30 | 90.23±3.02 | **92.05±2.08** | 93.01±2.33 | $\langle 0.75\rangle$ | 0.76 |
| Lung | **94.50±0.34** | 93.68±0.87 | 91.30±1.35 | 95.40±0.97 | $\langle 0.40\rangle$ | -0.08 |
| Lung_dis | **86.99±1.72** | 85.84±1.29 | N/A | 87.21±1.29 | $\langle 0.15\rangle$ | -0.22 |
| Lymphoma | **86.81±3.05** | 86.28±3.15 | 86.80±2.18 | 89.41±2.78 | $\langle 0.45\rangle$ | 0.35 |
| Nci9 | 54.17±2.10 | **55.28±3.90** | 49.17±3.57 | 53.89±0.79 | $\langle 0.10\rangle$ | 0.28 |
| GLIOMA | 75.00±1.53 | **78.00±4.62** | 69.33±0.94 | 76.33±2.69 | $\langle 0.20\rangle$ | -0.33 |
| colon | **82.53±2.54** | 80.38±2.54 | 81.99±1.45 | 83.06±1.23 | $\langle 0.70\rangle$ | 0.27 |
| ORL | **94.63±1.64** | 94.38±1.43 | 91.42±3.08 | 94.42±1.54 | $\langle 0.10\rangle$ | 0.21 |
| Yale | 69.29±5.87 | **70.10±4.68** | 66.16±3.76 | 71.41±3.45 | $\langle 0.55\rangle$ | -0.20 |
| GISETTE | **73.08±6.70** | 61.67±4.06 | 69.40±6.75 | 72.98±2.78 | $\langle 0.25\rangle$ | 0.10 |

$\overline{acc}\pm std$, SIOFS outperforms the second-best by 4.93% on the severe inter-class intrusion dataset GISETTE.

### 4.3. Ablation Study

**Influence of different class centers for SIOFS.** We only replace the proposed RDM center in SIOFS with CFDP and mean centers, respectively. Since CFDP and mean centers are parameter-free, for a fair discussion, we experimentally assign $\alpha = 0.1$ to the RDM center.

For Dataset #1∼#12 represented by low-level features, the corresponding $\overline{acc}\pm std$ results for SIOFS with RDM ($\alpha = 0.1$), CFDP, and mean centers are presented in the left 4 columns of Table 2. It can be seen that when using the RDM center with $\alpha = 0.1$, the performance of the features selected by SIOFS achieves the highest $\overline{acc}$ on eight datasets, and the second-highest $\overline{acc}$ on the rest. Notably, on TOX, the $\overline{acc}$ result for RDM center (79.82%) is 3.31% higher than the second-best CFDP center. On GISETTE, the $\overline{acc}$ for RDM center (73.08%) exceeds the second-best mean center by 3.68%. These results demonstrate the importance and effectiveness of the idea that averaging high-density instances reflects the overall characteristics of class body.

**Effectiveness of skewness coefficient.** We further evaluate

*Figure 5.* Performance of $\alpha$ for SIOFS on the 5 representative datasets. $m$: Number of selected features.

the effectiveness of SC by comparing the $\overline{acc} \pm std$ results. Based on (5), the coefficient $2 - \frac{\hat{s}^{(l)}}{3}$ is only simply replaced by "2" (denoted as "w/o SC"). The parameter $\alpha$ for the RDM center is still turned over $0.05, 0.1, \ldots, 0.95$.

The highest $\overline{acc}$ results and corresponding $\alpha$ values on Dataset #1∼#12 are reported in Table 2, under "w/o SC" and $\langle \alpha \rangle$. The $\overline{acc}$ differences between SIOFS and w/o SC are shown in Table 2, under "SIOFS -w/o SC". We observe that SIOFS achieves better classification performance on CLL, TOX, and Carcinom. For the other datasets, the $\overline{acc}$ differences are within $(-0.5\%, +0.5\%)$. In addition, the standard deviation of the 12 $\alpha$ values for SIOFS in row "$\langle \alpha \rangle$" of Table 1 is 0.18, while 0.22 for "w/o SC" in column "$\langle \alpha \rangle$" of Table 2. As analyzed in Sec. 1, both CLL and TOX contain classes with notable multiple high-density clusters (see Fig. 1d and 1e). To sum up, for datasets with intra-class ADMHC, SIOFS yields more robust results than w/o SC, and the SC is beneficial in regulating the threshold for obtaining the class body.

**An extension: comparing parameter-free SIOFS with IOFS.** We freeze $\alpha = 0.1$ as the parameter-free SIOFS. Comparing the row "IOFS" of Table 1 and the column "RDM $\alpha = 0.1$" of Table 2, we observe that $\overline{acc}$s increase on 6 out of 12 datasets and decrease on the rest. The maximum increment is 4.93% on GISETTE, while the maximum loss is 2.02% on Yale. The 6 datasets with reduced $\overline{acc}$ share a common characteristic that the distributions of class instances are relatively scattered. Consequently, as $\alpha$ changes, the RDM center has a larger drift, causing inappropriate representations of intra-class instance characteristics. On Dataset #1∼#12, the sum of all $\overline{acc}$s is 954.67 (%) for SIOFS with $\alpha = 0.1$ and 948.24 (%) for IOFS, indicating that the proposed SIOFS generally outperforms IOFS. Importantly, as analyzed in Sec. 1, the CLL dataset has distinct intra-class multiple high-density clusters, and GISETTE has severe intra-class asymmetric instance distribution. The parameter-free SIOFS significantly outperforms IOFS on both datasets. These imply that SIOFS is more applicable for high-dimensional data classification with strong intra-class ADMHC.

### 4.4. More Experimental Results and Analyses

**Challenge 3: FS on deep features.** Applying FS on deep learning-based cues is a trend in data classification (Lee

*Table 3.* Best global ACC (%) and NMI with corresponding number of selected features (in $(\cdot)$) obtained by FS methods on UCM, AID and ModelNet, under the top 5%, 10%, . . . , 95% of all features. AllFeat: All features are selected.

| Baseline | ACC | | | NMI | | |
|---|---|---|---|---|---|---|
| | UCM | AID | ModelNet | UCM | AID | ModelNet |
| AllFeat | 94.48(2048) | 84.36(2048) | 92.71(2048) | 0.720(2048) | 0.522(2048) | 0.797(2048) |
| Fisher | 94.38(1946) | 84.92(1434) | 92.79(1638) | 0.745(1126) | 0.552(1434) | 0.809(1946) |
| QMI | 94.33(1741) | 84.42(1741) | 92.91(922) | 0.742(1536) | 0.550(1229) | 0.824(1843) |
| ReliefF | 94.43(1843) | 84.80(1434) | 92.95(819) | 0.729(1741) | 0.548(1742) | **0.836**(614) |
| TRC | 94.48(1741) | 85.05(1434) | 92.83(1638) | 0.755(1638) | 0.559(1638) | 0.811(1946) |
| ILFS | 94.48(1741) | 83.93(1434) | **93.03**(410) | 0.731(1638) | 0.560(410) | 0.813(614) |
| FSDOC | 94.62(819) | 84.51(1434) | 92.91(1946) | 0.737(1126) | 0.552(1126) | $\underline{0.828}$(614) |
| FSTU | 94.76(1946) | 84.51(1946) | 92.67(1843) | 0.748(717) | 0.557(1536) | 0.808(1434) |
| FSNS | 94.67(1946) | 84.52(1946) | 92.63(1946) | 0.749(717) | 0.558(1536) | 0.806(1434) |
| ReOLSR | $\underline{94.81}$(1536) | 84.48(1946) | 92.83(512) | 0.723(1843) | 0.541(1946) | 0.808(1946) |
| MRMSR | 94.67(1024) | 84.64(410) | **93.03**(512) | 0.730(1434) | 0.550(1843) | 0.811(1843) |
| S$^2$DFS | 94.81(1741) | 84.35(1843) | 92.59(1843) | 0.735(1126) | 0.545(1536) | 0.824(717) |
| IOFS | 94.33(1741) | **85.83**(922) | 92.71(1946) | **0.778**(1331) | **0.594**(614) | 0.807(1946) |
| SIOFS | **94.95**(1946) | **85.83**(1024) | **93.03**(1843) | $\underline{0.775}$(1638) | $\underline{0.589}$(512) | 0.815(1434) |
| $\langle \alpha \rangle$ | $\langle 0.60 \rangle$ | $\langle 0.05 \rangle$ | $\langle 0.80 \rangle$ | $\langle 0.60 \rangle$ | $\langle 0.05 \rangle$ | $\langle 0.80 \rangle$ |
| w/o SC | $\underline{94.81}$(1536) | $\underline{85.80}$(1126) | $\underline{92.95}$(1946) | 0.769(1434) | 0.583(614) | 0.812(1946) |
| $\langle \alpha \rangle$ | $\langle 0.30 \rangle$ | $\langle 0.15 \rangle$ | $\langle 0.80 \rangle$ | $\langle 0.30 \rangle$ | $\langle 0.15 \rangle$ | $\langle 0.80 \rangle$ |

*Table 4.* Best global ACC (%) on Caltech 101 dataset under the top 60%, 70%, 80%, and 90% of all features. MRMSR fails to obtain the results within a limited time.

| | Fisher | QMI | ReliefF | TRC | ILFS | FSDOC | FSTU | FSNS | ReOLSR | S$^2$DFS | IOFS | SIOFS $\alpha=0.1$ |
|---|---|---|---|---|---|---|---|---|---|---|---|---|
| 60% | 40.79 | 39.7 | 38.58 | 37.76 | 42.38 | 43.56 | 40.63 | 40.63 | 40.99 | 42.38 | 45.45 | **45.74** |
| 70% | 41.68 | 40.92 | 40.36 | 39.11 | 42.38 | 43.43 | 43.47 | 43.47 | 41.45 | 43.43 | 48.84 | **48.98** |
| 80% | 42.61 | 42.57 | 41.88 | 40.63 | 42.48 | 43.93 | 42.77 | 42.77 | 42.67 | 43.56 | 51.52 | **51.58** |
| 90% | 43.43 | 43.23 | 42.90 | 41.19 | 43.60 | 43.20 | 42.44 | 42.44 | 42.77 | 43.66 | 53.83 | **54.13** |

et al., 2022). The challenges in aerial image classification (UCM, AID) and 3D object recognition (ModelNet) tasks arise from high inter-class intrusion (Xia et al., 2017).

Following the experimental protocol, Table 3 shows the best global ACC and NMI results of the same features selected by FS methods on UCM, AID, and ModelNet datasets. Some of the CMs are shown in Fig. 4. It can be seen that SIOFS achieves the best ACC results on these three datasets. Although the best NMI results for all datasets are not achieved by SIOFS, they are only slightly lower than the best. By comparing "SIOFS" and "w/o SC" in Table 3, we conclude that SC is also beneficial for FS on deep features.

**Challenge 4: FS on large-class-number dataset.** To validate the performance of SIOFS on large-class-number data, we use the Caltech101 dataset (Fei-Fei et al., 2004). This dataset is chosen also for its asymmetric intra-class instance distribution. Following the iterature (Chatfield et al., 2011; Yuan et al., 2022), Fisher Vector (FV) is employed to de-

*Table 5.* Time complexity and runtime (seconds). $\tau$ is the number of iterations of algorithms. $m$ is the number of selected features. "N/A": time complexity is not provided by original authors.

| Baseline | Time Complexity | Runtime (Seconds) | | | |
|---|---|---|---|---|---|
| | | CLL | GISETTE | Yale | AID |
| Fisher | $\mathcal{O}(dn)$ | 0.22 | 0.74 | 0.08 | 0.65 |
| QMI | N/A | 0.19 | 1.29 | 0.08 | 2.10 |
| ReliefF | $\mathcal{O}(dn)$ | 1.50 | 1323.96 | 0.38 | 1589.22 |
| TRC | $\mathcal{O}(dn^2+d^2n)$ | 4.63 | 271.22 | 0.07 | 329.09 |
| ILFS | $\mathcal{O}(n^{2.37}+\tau d+n+c)$ | 67.54 | 61.66 | 0.40 | 93.33 |
| FSDOC | $\mathcal{O}(dn+d^2)$ | 5.04 | 226.21 | 0.42 | 207.80 |
| FSTU | N/A | 3.25 | 13.07 | 0.81 | 100.91 |
| FSNS | N/A | 1.93 | 57.36 | 0.61 | 159.69 |
| ReOLSR | N/A | 10422.36 | 7960.63 | 67.13 | 4383.46 |
| S$^2$DFS | $\mathcal{O}(m^2d+nd^2)$ | 8129.74 | 1785.38 | 11.72 | 973.72 |
| IOFS | $\mathcal{O}(n^2+nc+c^2+dc)$ | 0.53 | 982.17 | 0.10 | 46.61 |
| SIOFS | $\mathcal{O}(n^2+n+nc+c^2+dc)$ | 0.92 | 865.58 | 0.10 | 259.60 |

scribe the images from this dataset. As FV is a high-level visual feature, and too much runtime should also be avoided, we conduct a systematic sampling (Harris & Stöcker, 1998) of the original FV features and select the top 60%, 70%, 80%, 90% of all features, and report the best global ACC result. $\alpha = 0.1$ is also frozen for SIOFS. As shown in Table 4, SIOFS outperforms the comparative methods in all cases. The corresponding information of Caltech101 dataset is given in Appendix F.

**Comparison with unsupervised FS methods.** The comparative methods and results are given in Appendix G, Table 7. SIOFS achieves the best or second-best performance on 15 datasets.

**Parameter sensitivity and runtime comparison.** Figure 5 illustrates the influence of $\alpha$ for SIOFS w.r.t. different numbers of selected features on 5 representative datasets, with the others in Appendix G. For some datasets, SIOFS is slightly sensitive to $\alpha$ when a few features are selected, but it is relatively robust in other cases. According to the definition of RDM center, $\alpha$ is the ratio of higher density to all instances in a class, thus $\alpha$ is affected by the aggregation behaviour of class instances. For some datasets, such as CLL (see Fig. 1b), due to the scattered distribution of class instances and multiple local high-density clusters, $\alpha$ has a distinct impact on the selected features.

Table 5 shows the time complexity and average runtime of 5 repeats for the FS methods on 4 representative datasets. The runtime of MRMSR is not reported due to its excessively long computation time. For larger datasets, such as GISETTE and AID, SIOFS takes more time. Objectively, most of the runtime for SIOFS is attributed to the $\mathcal{O}(n^2)$ calculation of the RDM center.

## 5. Conclusion

In this work, we propose the SIOFS method for high-dimensional data classification with intra-class ADMHC, where feature ranking is determined by the intrusion degrees

of feature on SIOs. The RDM center is proposed to characterize the class body. By measuring the distances from the class instances to the center, the modified SC is regulated and fused into the $3\sigma$ principle to define the class body. The intrusion degree is modeled based on the widely adopted conclusion of two intersecting spheres. Mathematical proofs or logical explanations for the key components of SIOFS are provided. Extensive experiments with 15 state-of-the-art FS methods are conducted on 16 multi-type datasets. Theoretical basis and experimental results demonstrate that, for data classification with intra-class ADMHC, evaluating features based on the intrusion degrees of features on SIOs is promising, and defining the class body by fusing the modified SC is appropriate. Theory and experiments support the superior performance and broad applicability of our method. Future work will focus on designing more powerful class centers and better mining the patterns of SIOs.

## Acknowledgements

This work was supported by the Fundamental Research Funds for the Central Universities under Grant B250201042, B250201046, and by Open Research Project of State Key Laboratory for Novel Software Technology, Nanjing University, under Grant KFKT2025B04. This work was also supported in part by the National Key R&D Program of China under Grant 2023YFC3006501, 2021YFB3900601, in part by the Natural Science Foundation of Jiangsu Province of China under Grant BK20242050, and in part by the Major Science and Technology Program of the Ministry of Water Resources of China under Grant SKS2022072. This work was supported in part by the High Performance Computing Platform, Hohai University.

## Impact Statement

The primary goal of this work is to advance the field of machine learning. SIOFS has the potential to improve analysis results in sectors that contain the classes with intra-class asymmetric instance distribution or multiple high-density clusters, from healthcare to finance. For instance, in healthcare, SIOFS can help identify critical health factors from longitudinal patient data, ensuring more accurate diagnoses and personalized treatment plans. Our work provides the theoretical foundation for further development of effective and interpretable models.

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

## A. Related Work

**FS for classification.** Most real-world classification problems require supervised learning, where the underling class probabilities and class-conditional probabilities are unknown, and each instance is associated with a class label (Tang et al., 2014; Theng & Bhoyar, 2024). In real-world cases, little knowledge about relevant features is available for learning. Therefore, more candidate features are introduced that are expected to better represent the instances, resulting in redundant features (Cai et al., 2018). For many classification tasks, it is challenging to learn discriminative classifiers before eliminating the redundant features. Meanwhile, this learning process is time-consuming. FS for classification aims to select a minimally sized subset of discriminative features according to certain criteria, improving feature interpretability and reducing the running time of learning models (Nie et al., 2019; Li et al., 2024).

**Review of FS.** FS methods can be categorized into three types: filter, wrapper and embedded methods (Wang et al., 2024; 2022). Filter methods rank feature importance based on the intrinsic characteristics of the data, using a predefined criterion, and are independent of the classifiers. As a result, they are computationally efficient and scalable for high-dimensional data. A typical filter algorithm consists of two steps (Tang et al., 2014): In the first step, features are ranked according to certain criteria; And in the second step, the highest-ranked features are selected for use in downstream learning tasks, such as classifiers. In wrapper methods, classifier performance is critical to FS. The performance of a specific classifier is optimized by searching for the best subset of features. Examining all possible subsets is an NP-hard problem, so a suboptimal solution is used, which remains costly for complex data classification tasks (Alelyani et al., 2013). Embedded methods bridge the gap between filter and wrapper methods by performing selection and classifier learning simultaneously (Jovic et al., 2015; Zhao et al., 2024). However, both wrapper and embedded method are prone to over-fitting because selection is part of the training process (Chen et al., 2022). The proposed SIOFS method is a supervised filter method.

## B. The Skewness Coefficient (SC)

According to the literature (Linton, 2017), the prototype of SC is

$$s^{(l)} = \frac{\frac{1}{n_l} \sum_{i=1}^{n_l} (d_i^{(l)} - \bar{X})^3}{\left(\frac{1}{n_l} \sum_{i=1}^{n_l} (d_i^{(l)} - \bar{X})^2\right)^{\frac{3}{2}}},$$

where $\bar{X}$ is the average of $d_1^{(l)}, d_2^{(l)}, \ldots, d_{n_l}^{(l)}$. $u^{(l)} = \text{RDM}(\{d_1^{(l)}, d_2^{(l)}, \ldots, d_{n_l}^{(l)}\}, 1) = \bar{X}$ when $\alpha = 1$, thus, we replace the $\bar{X}$ with $u^{(l)}$. That is, the term $(d_i^{(l)} - u^{(l)})^3$ in Equation (4) is directly derived from the term $(d_i^{(l)} - \bar{X})^3$. The SC is a statistical measure that quantifies the asymmetry of a data distribution. It indicates the degree to which data deviate from a symmetric, bell-shaped normal distribution.

One property of $s^{(l)}$ is that when $n_l$ is large enough and $\varepsilon > 0$, the frequency of $mode^{(l)} - \varepsilon$ is less than the frequency of $mode + \varepsilon$ if $s^{(l)} > 0$ (see Fig 6a and 6b), and the frequency of $mode^{(l)} - \varepsilon$ is greater than the frequency of $mode^{(l)} + \varepsilon$ if $s^{(l)} < 0$.

## C. Proofs

We begin by proving Theorem 1 from Section 3.1. We reiterate the theorems for completeness.

**Theorem 1.** *Let $\xi$ be the distance between the instance and the center in class $l$, $\xi$ is a continuous random variable. $d_1^{(l)}, d_2^{(l)}, \ldots, d_{n_l}^{(l)}$ is a random sample of $\xi$. The meanings of $u^{(l)}$ and $\hat{\sigma}^{(l)}$ are given in (4). When $\hat{\sigma}^{(l)} > 0$, we have the probability $Pr(\xi < u^{(l)} + 2\hat{\sigma}^{(l)}) > \frac{3}{4}$.*

*Proof.* Suppose the probability density function of $\xi$ is $f(x)$, based on the property of probability density function (Harris & Stöcker, 1998), we have $f(x) \geq 0$.

When $x \geq u^{(l)} + 2\hat{\sigma}^{(l)}$, i.e., $x - u^{(l)} \geq 2\hat{\sigma}^{(l)} > 0$, we have

$$f(x) \leq \frac{(x - u^{(l)})^2}{4(\hat{\sigma}^{(l)})^2} f(x).$$

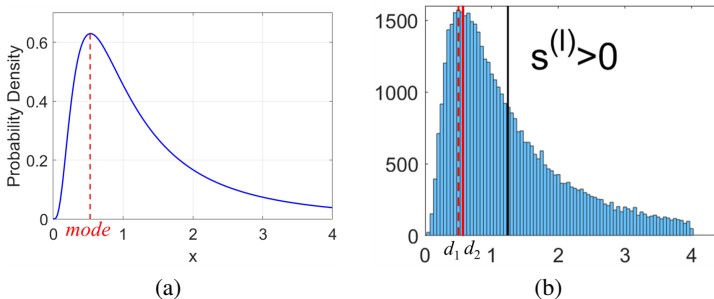

*Figure 6.* (a) A probability density function of skewed distribution. (b) Histograms of 100,000 data sampled from the distribution in (a).

Then,

$$Pr(\xi < \mathrm{u}^{(l)} + 2\hat{\sigma}^{(l)}) = 1 - \int_{\mathrm{u}^{(l)}+2\hat{\sigma}^{(l)}}^{+\infty} f(x)dx$$

$$\geq 1 - \int_{\mathrm{u}^{(l)}+2\hat{\sigma}^{(l)}}^{+\infty} \frac{(x - \mathrm{u}^{(l)})^2}{4(\hat{\sigma}^{(l)})^2} f(x)dx$$

$$> 1 - \frac{1}{4(\hat{\sigma}^{(l)})^2} \int_{-\infty}^{+\infty} (x - \mathrm{u}^{(l)})^2 f(x)dx.$$

Here, $\int_{-\infty}^{+\infty}(x - \mathrm{u}^{(l)})^2 f(x)dx$ is the variance w.r.t $\mathrm{u}^{(l)}$ and $(\hat{\sigma}^{(l)})^2$ is a precise estimation of $\int_{-\infty}^{+\infty}(x - \mathrm{u}^{(l)})^2 f(x)dx$. Thus, we have $Pr(\xi < \mathrm{u}^{(l)} + 2\hat{\sigma}^{(l)}) > \frac{3}{4}$. $\qquad\square$

**Theorem 2.** *For $d_1^{(l)}, d_2^{(l)}, \ldots, d_{n_l}^{(l)}$, $\mathrm{u}^{(l)}, \hat{s}^{(l)}$ are the same as in* (4)*, and $mode^{(l)}$ is the same as the footnote in Section 3.1. When $\alpha \in (0, 1]$, $mode^{(l)} \leq \mathrm{u}^{(l)} \leq average^{(l)}$ holds with probability 1 if $\hat{s}^{(l)} > 0$ and $average^{(l)} \leq \mathrm{u}^{(l)} \leq mode^{(l)}$ holds with probability 1 if $\hat{s}^{(l)} < 0$.*

*Proof.* Assume $\hat{s}^{(l)} > 0$. According to the definition of RDM center, $\mathrm{u}^{(l)}$ is the average of the top $\lceil \alpha \cdot n_l \rceil$ high density values in $d_1^{(l)}, d_2^{(l)}, \ldots, d_{n_l}^{(l)}$. When $\alpha$ is smaller in $(0, 1]$ but $\lceil \alpha \cdot n_l \rceil = 1$, we have $\mathrm{u}^{(l)} = mode^{(l)}$. For $d_1^{(l)}, d_2^{(l)}, \ldots, d_{n_l}^{(l)}$, let $d_1 = mode^{(l)}$ and $d_2, d_3$ denote the 2nd and 3rd highest density values, respectively. $\hat{s}^{(l)}$ has the same property of $s^{(l)}$. When $\alpha$ is smaller in $(0, 1]$ but $\lceil \alpha \cdot n_l \rceil = 2$, based on the property of $s^{(l)}$ (see Appendix B), and considering that $n_l$ is often not large enough, at least, $d_2 > d_1$ holds with probability 1. Then $\mathrm{u}^{(l)} = \frac{d_1+d_2}{2} > d_1$ with probability 1. When $\alpha$ is smaller in $(0, 1]$ but $\lceil \alpha \cdot n_l \rceil = 3$, if $d_3 > d_1$, then $\mathrm{u}^{(l)} = \frac{1}{3}(d_1 + d_2 + d_3) > d_1$; if $d_3 < d_1$ (see Fig. 6b), $d_3$ can only be very close to $d_1$, namely, $d_1 - d_3 \leq d_2 - d_1$, $\mathrm{u}^{(l)} = \frac{1}{3}(d_1 + d_2 + d_3) \geq d_1$ holds with probability 1 at least. When $\lceil \alpha \cdot n_l \rceil = 4, 5, \ldots$, the same conclusion can be obtained.

When $\hat{s}^{(l)} > 0$, the smaller values in $d_1^{(l)}, d_2^{(l)}, \ldots, d_{n_l}^{(l)}$ are denser. Based on the difinition of RDM center, if $\hat{s}^{(l)} > 0$ and $\alpha < 1$, after the largest $\lceil (1 - \alpha) \cdot n_l \rceil$ values in $d_1^{(l)}, d_2^{(l)}, \ldots, d_{n_l}^{(l)}$ are removed, the remainder is averaged as the $\mathrm{u}^{(l)}$. As we know, the average of a set of numbers with large values removed is less than the average of all. Meanwhile, $\mathrm{u}^{(l)}$ is the average of all $d_1^{(l)}, d_2^{(l)}, \ldots, d_{n_l}^{(l)}$ when $\alpha = 1$. Consequently, if $\hat{s}^{(l)} > 0$, as $\alpha$ decreases from 1, $\mathrm{u}^{(l)}$ *gradually gets smaller from the average of all $d_1^{(l)}, d_2^{(l)}, \ldots, d_{n_l}^{(l)}$*. Combined with the conclusion that $mode < average$ if $s^{(l)} > 0$, we have $mode^{(l)} \leq \mathrm{u}^{(l)} \leq average^{(l)}$ when $\alpha$ is the larger value in $(0, 1]$. In conclusion, when $\alpha \in (0, 1]$, we have $mode^{(l)} \leq \mathrm{u}^{(l)} \leq average^{(l)}$ holds with probability 1 if $\hat{s}^{(l)} > 0$.

Similarly, when $\alpha \in (0, 1]$, we have $average^{(l)} \leq \mathrm{u}^{(l)} \leq mode^{(l)}$ if $\hat{s}^{(l)} < 0$. $\qquad\square$

# D. Theoretical Foundation for Rationality of Evaluation Criterion

For every $\mathbf{x}_i^{(k)} \in \mathcal{X}_o^{(kl_0)}$, we have $\sum_{f=1}^{d} S_{if}^{(kl_0)} > 0$ due to (10), and thus $\sum_{f=1}^{d} \bar{S}_f^{(kl_0)} > 0$. However, for feature $f = 1, 2, \ldots, d$, $\bar{S}_f^{(kl_0)}$ can be positive or negative. There must be at least one $f \in \{1, 2, \ldots, d\}$ that meets $\bar{S}_f^{(kl_0)} > 0$. Based on our evaluation

---

**Algorithm 1** Obtaining the RDM center of class $k$ ($n_k \geq 3$).

---

**Input:** Data set $\{\mathbf{x}_1^{(k)}, \mathbf{x}_2^{(k)}, \ldots, \mathbf{x}_{n_k}^{(k)}\}$, parameter $\alpha$;
**Output:** The RDM center $\mathbf{u}^{(k)} \in \mathbb{R}^d$;
  1: Initialize the high density instance subset $\mathcal{H}^{(k)} = \emptyset$;
  2: Compute $d_{ij}^{(k)} = \|\mathbf{x}_i^{(k)} - \mathbf{x}_j^{(k)}\|_1$ for all pairwise $\mathbf{x}_i^{(k)}$ and $\mathbf{x}_j^{(k)}$, $i, j = 1, 2, \ldots, n_k$;
  3: **for** $i = 1, 2, \ldots, n_k$ **do**
  4:     Compute $\varepsilon_i^{(k)} = \text{median}(d_{i1}^{(k)}, d_{i2}^{(k)}, \ldots, d_{in_k}^{(k)})$;
  5: **end for**
  6: Compute $T^{(k)}$ via (2);
  7: **for** $i = 1, 2, \ldots, n_k$ **do**
  8:     Assign $\mathbf{x}_i^{(k)} \to \mathcal{H}^{(k)}$ if $\varepsilon_i^{(k)} \leq T^{(k)}$;
  9: **end for**
10: Compute $\mathbf{u}^{(k)}$ via (3).

---

**Algorithm 2** Obtaining the SIOs set $\mathcal{X}_o^{(kl_0)}$ for class $k$ towards $l_0$ ($l_0 \neq k$)

---

**Input:** $\{\mathbf{x}_1^{(k)}, \mathbf{x}_2^{(k)}, \ldots, \mathbf{x}_{n_k}^{(k)}\}$, $\mathbf{u}^{(k)}$, $\Theta^{(k)}$, $k = 1, 2, \ldots, c$;
**Output:** $\mathcal{X}_o^{(kl_0)}$;
  1: **for** $k = 1, 2, \ldots, c$ **do**
  2:     Acquire $\mathcal{X}^{(kl_0)}$ for the class pair $k, l_0$ via (6) and $l_0 = \arg\min_{l \in \{l_1, l_2, \ldots\}} (\|\mathbf{x}_i^{(k)} - \mathbf{u}^{(l)}\|_1)$;
  3:     Acquire $\mathcal{X}^{(l_0)k} = \{\mathbf{x}_i^{(l_0)} : \|\mathbf{x}_i^{(l_0)} - \mathbf{u}^{(k)}\|_1 < \Theta^{(k)}, i = 1, 2, \ldots, n_{l_0}\}$;
  4:     **if** $\mathcal{X}^{(kl_0)} \neq \emptyset$ and $\mathcal{X}^{(l_0)k} \neq \emptyset$ **then**
  5:         Compute $D^{(l_0)k}$ via (7);
  6:         Acquire $\mathcal{X}_o^{(kl_0)}$, where $\mathbf{x}_i^{(k)} \in \mathcal{X}_o^{(kl_0)}$ meets (8).
  7:     **end if**
  8: **end for**

---

**Algorithm 3** Final Feature selection of SIOFS method

---

**Input:** $\mathbf{x}_i^{(k)} \in \mathcal{X}_o^{(kl_0)}$, $k = 1, 2, \ldots, c$, the number $m$ of selected features;
**Output:** The selected $m$ features;
  1: **while** $N_{ip}$ **do**
  2:     **for** $f = 1, 2, \ldots, d$ **do**
  3:         Compute $D_f^{(l_0)k} = \frac{1}{|\mathcal{X}^{(l_0)k}|} \sum_{\mathbf{x}^{(l_0)} \in \mathcal{X}^{(l_0)k}} |x_{if}^{(l_0)} - u_f^{(l_0)}|$ and $S_{if}^{(kl_0)} = |x_{if}^{(k)} - u_f^{(k)}| + D_f^{(l_0)k} - |u_f^{(k)} - u_f^{(l_0)}|$;
  4:         Compute $\bar{S}_f^{(kl_0)} = \frac{1}{|\mathcal{X}_o^{(kl_0)}|} \sum_{\mathbf{x}_i^{(k)} \in \mathcal{X}_o^{(kl_0)}} S_{if}^{(kl_0)}$;
  5:     **end for**
  6: **end while**
  7: Acquire $\mathbf{P} = (P_{ij})_{\lceil 0.5 N_{ip} \times d \rceil}$ whose row is the vector $(\bar{S}_1^{(kl_0)}, \bar{S}_2^{(kl_0)}, \ldots, \bar{S}_d^{(kl_0)})$;
  8: Compute $(s_1, s_2, \ldots, s_d)$ via (11);
  9: Sort $s_1, s_2, \ldots, s_d$ in ascending order and select the features corresponding to the first $m$ values.

---

criterion, the first *unselected* feature $f_0$ must satisfy $\bar{S}_{f_0}^{(kl_0)} > 0$. Thus, $\sum_{f \in \{1,2,\ldots,d\} - \{f_0\}} \bar{S}_f^{(kl_0)} < \sum_{f=1}^d \bar{S}_f^{(kl_0)}$. That is, the intrusion degree between class $k$ and $l_0$ is descending in the feature space $\{1, 2, \ldots, d\} - \{f_0\}$ than $\{1, 2, \ldots, d\}$.

## E. Algorithms and Time Complexity Analysis

**Obtaining the RDM center.** Corresponding procedure is shown in Algorithm 1. For class $k$ ($k = 1, 2, \ldots, c$), the time complexity of *Step 2* is $\mathcal{O}(\frac{1}{2} n_k^2)$ due to the symmetry matrix $(d_{ij})_{n_k \times n_k}$, and computing *Step 4* and *Step 6* cost $\mathcal{O}(n_k)$ and $n_k$ time, respectively. Thus, the total time of Algorithm 1 is $\mathcal{O}(\sum_{k=1}^c (\frac{1}{2} n_k^2 + 2n_k))$. Due to $\sum_{k=1}^c n_k^2 \leq (\sum_{k=1}^c n_k)^2$, the time complexity of Algorithm 1 limits to $\mathcal{O}(n^2 + n)$.

**Procedure of obtaining the SIOs set and the final feature selection.** The procedure of obtaining the SIOs set $\mathcal{X}_o^{(kl_0)}$ and the final feature selection are presented in Algorithm 2 and Algorithm 3, respectively. Let $m$ denote the number of selected features. The notations $n, d, c$ have the same meanings as them in Section 2.

- Algorithm 2: For $k = 1, 2, \ldots, c$, computing *Step* 2∼7 cost $\mathcal{O}(nc)$ time for $\|\mathbf{x}_i^{(k)} - \mathbf{u}^{(k)}\|_1$, and $\mathcal{O}(c^2)$ time for $\|\mathbf{u}^{(k)} - \mathbf{u}^{(l_0)}\|_1$. Thus, the total time of Algorithm 2 is $\mathcal{O}(nc + c^2)$.

- Algorithm 3: We assume that $|\mathcal{X}_o^{(kl_0)}| \ll n_k$ and all classes contain the SIOs, i.e., $N_{ip} = c$. Computing *Steps* 1∼6 costs $\mathcal{O}(dc)$ time. Computing $\mathbf{P}$ costs $\mathcal{O}(dc)$. Thus, the total time of Algorithm 3 is $\mathcal{O}(dc)$.

Combining the time of Algorithm 1, the total time of SIOFS is $\mathcal{O}(n^2 + n + d)$ for small-sized high-dimensional datasets due to $c \ll d$, and the upper bound of the time complexity of SIOFS is $\mathcal{O}(n^2 + n + nc + c^2 + dc)$ for large-scale datasets. Objectively, the total time of SIOFS is mainly due to Algorithm 1.

## F. Additional Experimental Settings

In this section, we describe the experimental details, including baseline methods, datasets and the extraction for deep features. All the experiments are performed on windows-7 operating system (Intel Xeon Gold 6128 CPU @ 3.40GHz 16.0GB RAM).

*Table 6.* Information of the baseline methods and datasets. The types of baselines are $f$=filter, $e$=embedded and $s$=supervised, $u$=unsupervised.

| Baseline | Ref. | Type | | | Dataset | Type | #Instances | #$d$ | #$c$ |
|---|---|---|---|---|---|---|---|---|---|
| Fisher | (Duda et al., 2001) | s | f | #1 | CLL | Biology | 111 | 11340 | 3 |
| QMI | (Zhang et al., 2016) | s | f | #2 | TOX | Biology | 171 | 5748 | 4 |
| ReliefF | (Robnik-Sikonja & Kononenko, 2003) | s | f | #3 | Carcinom | Biology | 174 | 9182 | 11 |
| TRC | (Nie et al., 2008) | s | f | #4 | Lung | Biology | 203 | 3312 | 5 |
| ILFS | (Roffo et al., 2017) | s | f | #5 | Lung_dis | Biology | 73 | 325 | 7 |
| FSDOC | (Yuan et al., 2022) | s | f | #6 | Lymphoma | Biology | 96 | 4026 | 9 |
| FSTU | (Lohrmann & Luukka, 2022) | s | f | #7 | Nci9 | Biology | 60 | 9712 | 9 |
| FSNS | (Lohrmann & Luukka, 2022) | s | f | #8 | GLIOMA | Biology | 50 | 4434 | 4 |
| ReOLSR | (Zhao et al., 2018) | s | f | #9 | colon | Biology | 62 | 2000 | 2 |
| MRMSR | (Wang et al., 2023) | s | f | #10 | ORL | Face Images | 400 | 1024 | 40 |
| S$^2$DFS | (Nie et al., 2022) | s | e | #11 | Yale | Face Images | 165 | 1024 | 15 |
| IOFS | (Yuan et al., 2024) | s | f | #12 | GISETTE | NIPS FS | 7000 | 5000 | 2 |
| InfFS$_U$ | (Roffo et al., 2021) | u | f | #13 | UCM (2010) | Aerial Images | 2100 | 2048 | 21 |
| EGCFS | (Zhang et al., 2022) | u | e | #14 | AID (2017) | Aerial Images | 10000 | 2048 | 30 |
| FSDK | (Nie et al., 2024) | u | e | #15 | ModelNet (2015) | 3D CAD Models | 12308 | 2048 | 40 |
| | | | | #16 | Caltech101 (2004) | Objects | 3030 | 5243 | 101 |

**Descriptions of baselines.** As shown in Table 6, twelve supervised methods and three unsupervised methods are considered for comparison. As a classical filter FS method, **Fisher** (Duda et al., 2001) method scores features as the ratio of inter-class variance and intra-class separation of data. **QMI** (Zhang et al., 2016) method uses the quantized discrete variables of data entropy for FS. **ReliefF** (Robnik-Sikonja & Kononenko, 2003; Kononenko, 1994) relies on instances and their neighbors to estimate feature quality, which is greedy to minimize the redundancy among the selected features. **TRC** (Nie et al., 2008) method proposes a trace ratio criterion to evaluate features. **ILFS** (Roffo et al., 2017) is an infinite latent FS method, which performs the ranking step by considering all the possible feature subsets. In **FSDOC** (Yuan et al., 2022) method, the characteristics of directional outliers are mined and employed for feature scoring. Supervised **FSTU** and **FSNS** (Lohrmann & Luukka, 2022) methods score features from a total uncertainty and non-specificity perspective of possibility theory, respectively, in the context of classification. **ReOLSR** (Zhao et al., 2018) method adopts the orthogonal constraint on the transformation matrix in least squares regression model to preserve more data structure information, and uses an iterative algorithm to transform the unbalanced orthogonal Procrustes problem into balanced one. In **MRMSR** (Wang et al., 2023) method, a criterion of max-relevance and min-supervised-redundancy is introduced using the normalized feature relevance metric and supervised similarity measure. **S$^2$DFS** (Wang et al., 2020; Nie et al., 2022) method uses the trace ratio formulated objective functions to ensure the discriminability of selected features. **IOFS** (Yuan et al., 2024) scores features based on the characteristics of a few outliers that are within the body of the class to which they do not belong. **InfFS$_U$** (Roffo et al., 2021) is an unsupervised filter method, where features are identified as nodes in a graph and the selection is a path through

these features. **EGCFS** (Zhang et al., 2022) method directly embeds graph learning into the optimization process. **FSDK** (Nie et al., 2024) is a fast sparse discriminative K-means method, where the weighted pseudo-label matrix with discrete trait is introduced to avoid trivial solution from unsupervised least-squares regression.

**Descriptions about datasets and difficulties for data classification.** Table 6 shows the dataset types, total numbers of all instances, feature dimensions and classes, respectively. Descriptions and difficulties for data classification are as follows.

- The first 11 datasets are in small-sized high-dimensional scenario, where the instances are difficult to collect and the number of measurements made on each instance can easily reach the order of thousands (e.g., set of DNA sequences). They bring challenges such as the curse of dimensionality for FS. These datasets are chosen for their variability in terms of the number of features (from 325 to 11340), characterizing 50 to 400 instances. Datasets are downloaded from `https://jundongl.github.io/scikit-feature/datasets.html`.

- GISETTE dataset is from the NIPS 2003 FS challenge, which has two severely confusable handwritten digits 4 and 9 extracted from MNIST data (LeCun et al., 1998) (see Fig. 1c). It is also downloaded from `https://jundongl.github.io/scikit-feature/datasets.html`.

- UCM (Yang & Newsam, 2010) dataset is publicly available on `http://vision.ucmerced.edu/datasets/landuse.html`. It consists of 21 classes of land-use images selected from aerial orthoimagery with the pixel resolution of 1 ft. The original images were downloaded from the United States Geological Survey National Map of the following U.S. regions.

- AID (Xia et al., 2017) is a large-scale dataset for aerial scene classification (see Fig. 1a). It has a number of 10,000 images within 30 classes. The numbers of sample images vary a lot with different aerial scene types from 230 up to 420. A main difference between AID and UCM datasets is that AID has multiresolutions that the pixel resolution changes from about 8 m to about half a meter, and thus, the size of each aerial image is fixed to be $600 \times 600$ pixels to cover a scene with various resolutions. AID has higher intra-class variations, smaller inter-class dissimilarity and a relatively large scale. The AID is downloaded from `https://captain-whu.github.io/AID/`.

- ModelNet (Wu et al., 2015) consists of 13,211 3D synthetic models for general objects, with 9843 training samples and 2468 testing samples ranged within 40 classes. We only use its training and test sets, which are downloaded from `https://modelnet.cs.princeton.edu/`.

- Caltech101 (Fei-Fei et al., 2004) is designed to advance research in multi-class object recognition and image classification. This dataset contains 101 object classes, including animals, vehicles, everyday objects. It also has high inter-class similarity, intra-class variability in pose, lighting, and scale, and the cluttered backgrounds in some classes. Caltech101 dataset is downloaded from `https://data.caltech.edu/records/mzrjq-6wc02`.

**Deep feature extraction.** To extract the deep features, we use a residual network with depth of 152 (He et al., 2016) pre-trained on ImageNet (Russakovsky et al., 2015) as the backbone for UCM and AID datasets. For ModelNet dataset, we use CurveNet (Xiang et al., 2021) as the backbone and use the input of the classification layer as the feature values of the instances. The dimensions of all these deep features are 2048. All implementation details are same as in literature (Yuan et al., 2024).

**Systematic sampling of FV features.** Following the settings in (Chatfield et al., 2011; Yuan et al., 2022), the dimension of the original FV feature is 262144. We conduct the systematic sampling (Harris & Stöcker, 1998) of FV feature, where the 1st, 51st, 101st, ..., features are sampled. The final dimension of the sampled FV is 5243.

## G. Additional Results and Analyses

**Additional ACC and NMI results.** The ACC and NMI results of the remaining datasets (CLL, Lung, Lymphoma, Nci9, GLIOMA, colon and Yale) are shown in Fig. 7.

**Comparisons with unsupervised FS methods.** Unsupervised FS methods select features by seeking the intrinsic structures of data without label information. We follow the same implementations as in *Challenge 1 and 3* for 15 dataset. Table 7 shows the results of SIOFS in comparison with three unsupervised FS methods (InfFS$_U$, EGCFS and FSDK). As shown in Table 7 that SIOFS achieves the best classification results on 13 out of 15 datasets and the second best on the rest. This also shows the superiority of SIOFS.

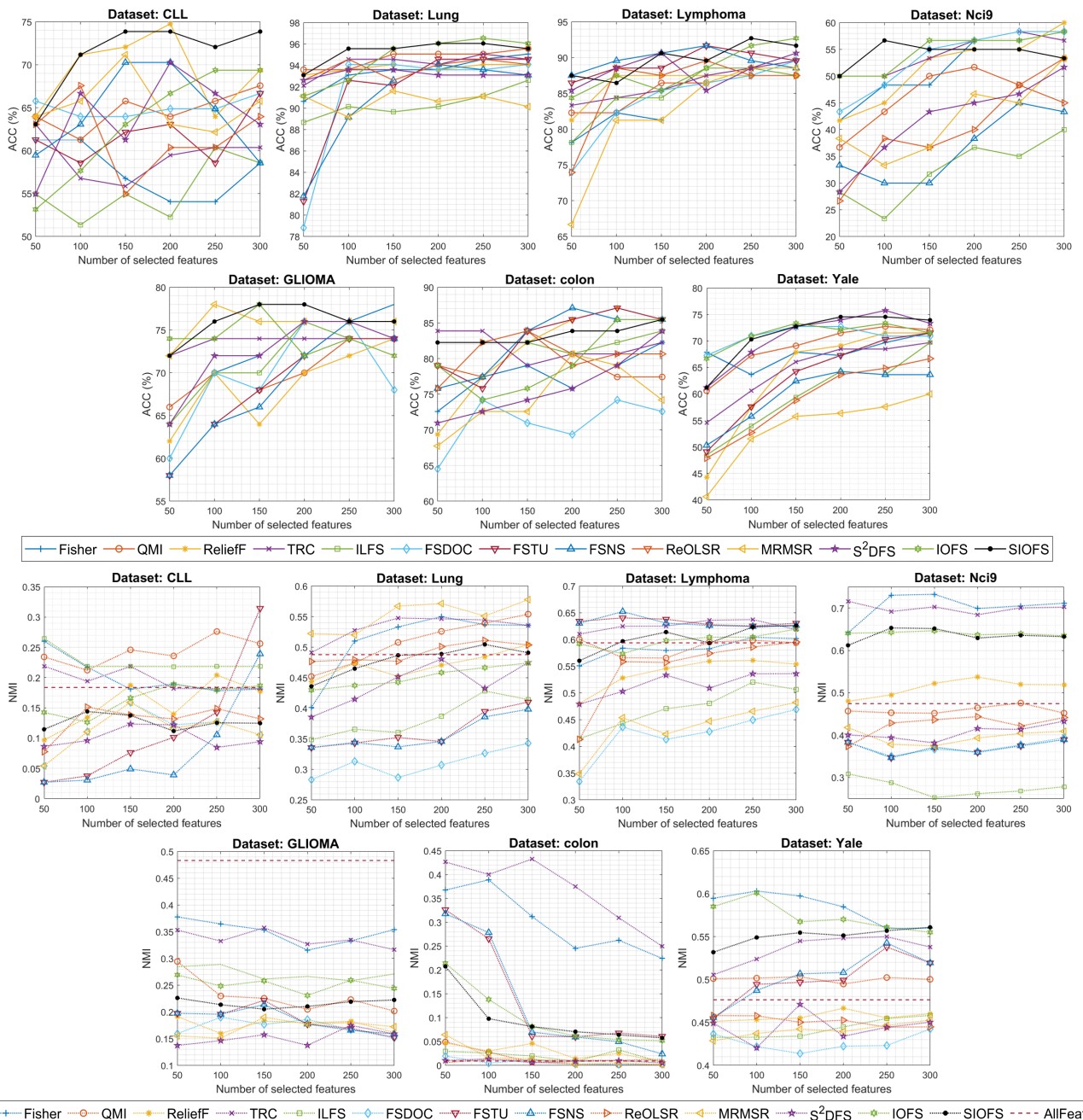

*Figure 7.* ACC and NMI results of FS methods on the remaining datasets w.r.t the top 50, 100, . . . , 300 features. AllFeat: All features are selected.

*Table 7.* Comparison of ACC (%) for unsupervised FS methods and SIOFS. The values enclosed in $\langle \cdot \rangle$ show the corresponding number of selected features.

| Dataset | InfFS$_U$ | EGCFS | FSDK | SIOFS |
|---|---|---|---|---|
| CLL | $62.76_{\pm 5.12}$ | $65.32_{\pm 5.09}$ | $59.46_{\pm 2.70}$ | $\mathbf{71.32}_{\pm \mathbf{3.84}}$ |
| TOX | $\mathbf{90.45}_{\pm \mathbf{4.13}}$ | $78.85_{\pm 6.36}$ | $85.28_{\pm 6.68}$ | $83.24_{\pm 4.99}$ |
| Carcinom | $93.49_{\pm 3.28}$ | $83.34_{\pm 6.34}$ | $90.13_{\pm 2.74}$ | $\mathbf{93.77}_{\pm \mathbf{2.72}}$ |
| Lung | $93.76_{\pm 1.09}$ | $92.29_{\pm 1.35}$ | $94.75_{\pm 1.01}$ | $\mathbf{95.32}_{\pm \mathbf{1.02}}$ |
| Lung_dis | $85.39_{\pm 1.29}$ | $84.93_{\pm 1.12}$ | $84.70_{\pm 2.43}$ | $\mathbf{86.99}_{\pm \mathbf{1.90}}$ |
| Lymphoma | $87.68_{\pm 3.26}$ | $88.37_{\pm 2.65}$ | $87.85_{\pm 1.43}$ | $\mathbf{89.76}_{\pm \mathbf{2.20}}$ |
| Nci9 | $54.45_{\pm 5.42}$ | $47.78_{\pm 5.75}$ | $43.06_{\pm 5.22}$ | $\mathbf{54.17}_{\pm \mathbf{2.10}}$ |
| GLIOMA | $62.33_{\pm 6.57}$ | $72.00_{\pm 3.83}$ | $74.67_{\pm 1.89}$ | $\mathbf{76.00}_{\pm \mathbf{2.00}}$ |
| colon | $79.30_{\pm 3.89}$ | $80.83_{\pm 6.99}$ | $77.42_{\pm 5.51}$ | $\mathbf{83.33}_{\pm \mathbf{1.20}}$ |
| ORL | $80.88_{\pm 9.45}$ | $\mathbf{95.13}_{\pm \mathbf{1.18}}$ | $93.92_{\pm 2.62}$ | $94.63_{\pm 1.64}$ |
| Yale | $47.88_{\pm 10.24}$ | $70.91_{\pm 4.30}$ | $64.04_{\pm 8.21}$ | $\mathbf{71.21}_{\pm \mathbf{4.70}}$ |
| GISETTE | $67.89_{\pm 5.54}$ | $61.56_{\pm 9.89}$ | $68.14_{\pm 6.11}$ | $\mathbf{73.08}_{\pm \mathbf{6.70}}$ |
| UCM | $94.62_{(1946)}$ | $94.71_{(1843)}$ | $94.48_{(1741)}$ | $\mathbf{94.95}_{(1946)}$ |
| AID | $83.94_{(1946)}$ | $84.40_{(1731)}$ | $84.27_{(1946)}$ | $\mathbf{85.83}_{(1024)}$ |
| ModelNet | $92.67_{(1946)}$ | $92.75_{(512)}$ | $92.71_{(614)}$ | $\mathbf{93.03}_{(1843)}$ |

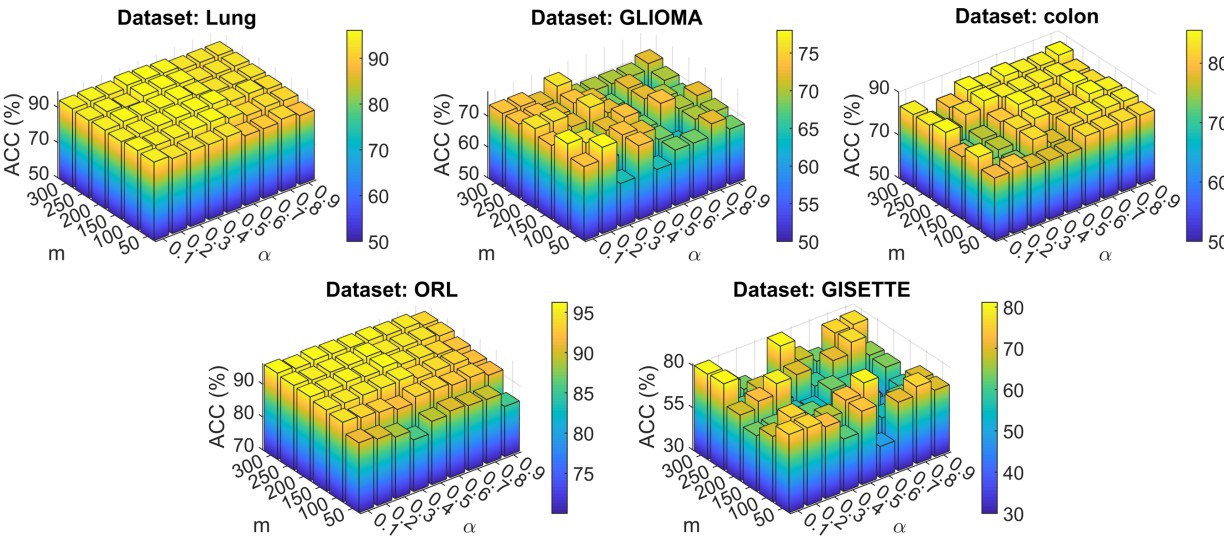

*Figure 8.* Performance of $\alpha$ for SIOFS on the rest 5 datasets. Since the SIOs can not be obtained with larger $\alpha$, results on Lung_dis and Lymphoma datasets are not reported.

