# OpenReview forum: "Stray Intrusive Outliers-Based Feature Selection on Intra-Class Asymmetric Instance Distribution or Multiple High-Density Clusters"
_ICML.cc/2025/Conference — ICML 2025 poster_

### Official Review · Reviewer_5k2J · 2025-03-13

**Overall Recommendation:** 3

**Summary:**

This paper proposes a supervised FS method, Stray Intrusive Outliers-based FS (SIOFS), for data classification with intra-class ADMHC. By focusing on Stray Intrusive Outliers (SIOs), SIOFS modifies the skewness coefficient and fuses the threshold in the 3σ principle to identify the class body, scoring features based on the intrusion degree of SIOs.

**Claims And Evidence:**

Please see Weaknesses.

**Essential References Not Discussed:**

None.

**Experimental Designs Or Analyses:**

Checked.

**Methods And Evaluation Criteria:**

Please see Weaknesses,

**Other Comments Or Suggestions:**

Please see Weaknesses.

**Other Strengths And Weaknesses:**

Strengths
1. The paper structure is clear and is easy to follow.
2. The proposed method is novel.
3. The method is supported by theoretical evidence and empirical evidence.

Weaknesses
1. At the third paragraph in Introduction, the authors mention that as shown in Fig. 1b, class “2” has two high-density clusters, but there is only class "1" and "3" in the figure.
2. What is definition of $3\sigma$ principle? Can authors explain its insight?
3. The explanation in Line 132 for equation 3 assumes that outliers have low instance density and thus will not be included. However, there are different types of outliers and low density assumption can not guarantee that all outliers are not included.
4. Why using l1 distance in equation 1?
5. The authors need to explain why the modified SC is formulated as the type of Eq.4. Why $(d_i^{(l)} - u^{(l)})^3$ is used? Is this mean that other terms like $(d_i^{(l)} - u^{(l)})^1$ can not be used?
6. In Line 192, why normalize $\hat{s}^{(l)}$ by $\hat{s}^{(l)}/3$?

**Questions For Authors:**

Please see Weaknesses.

**Relation To Broader Scientific Literature:**

Most current FS methods score features based on the characteristics of all training instances. Existing FS methods rarely aim to identify the class body in the context of intra-class multiple high-density clusters. This paper addresses them.

**Theoretical Claims:**

Please see Weaknesses.

---

> ### Author Rebuttal · Authors · 2025-03-31
>
> We sincerely thank Reviewer 5k2j for the constructive and valuable comments. The concerns are addressed as follows.
> ## Q1: In Fig. 1b, class "2" has two high-density clusters, but there is only class "1" and "3" in the figure.
> Many thanks for the comment. We revise this typo from class "2" to class "3".
> ## Q2: What is the definition of 3σ principle? Can authors explain its insight?
> Sorry for the unclear description. We add the classical literature (Harris & Stocker, 1998) and **[1]** to support the 3$\sigma$ principle. For $\xi\sim N(\mu,\sigma^2)$, the probability (denoted as $Pr$) that $Pr(\xi\in(\mu-3\sigma,\mu+3\sigma))>0.997$. This provides a robust framework for analysing variability in normally distributed data and guides thresholds for outlier detection in the context of ADMHC.
>
> **[1]** Groeneveld, R. A., & Meeden, G. (1984). Measuring Skewness and Kurtosis. Journal of the Royal Statistical Society: Series D (The Statistician), 33(4), 391-399.
> ## Q3: There are different types of outliers and low density assumption can not guarantee that all outliers are not included.
> We clarify that due to random errors in the data, especially in synthetic datasets, it is impossible to correctly identify all of them. In this paper, only low-density cases are identified and used as outliers. We will include the above details of the outliers we have identified in the final version.
> ## Q4: Why using l1 distance in Eq.(1)?
> As presented in the original paper (see line 79, right column), the $\ell_1$ norm can treat each component of the feature vector equally. $\ell_1$ treats all $\vert x_{if}^{(k)}-x_{jf}^{(k)} \vert$ linearly while $\ell_2$ penalizes large $\vert x_{if}^{(k)}-x_{jf}^{(k)}\vert$ quadratically. This makes it easier to identify outliers in $\ell_1$ than in $\ell_2$.
>
> As suggested, we add comparisons with $\ell_1$ distance and the typical $\ell_2$ distance. Note that we only replace $\ell_1$ distance with $\ell_2$ distance. As shown in **Rebuttal Table D**, SIOFS with $\ell_1$ distance outperforms that with $\ell_2$ distance on all datasets. We will include these details in the final version.
>
> **Rebuttal Table D.** Comparative results of SIOFS with $\ell_1$ distance (denoted as "SIOFS") and $\ell_2$ distances (denoted as "w/ $\ell_2$") on some datasets. "w/ $(\cdot)^1$" means that the $(\cdot)^3$ terms are directly replaced by $(\cdot)^1$ in Eq.(4). The average ACC over all 11 datasets is also reported for a comprehensive comparison.
> |ACC (%) $\uparrow$|SIOFS|w/ $\ell_2$|w/ $(\cdot)^1$|
> |-|-|-|-|
> |CLL|**71.32**$\pm$3.84|65.47$\pm$2.58|70.57$\pm$1.85|
> |TOX|**83.24**$\pm$4.99|81.19$\pm$2.93|**83.24**$\pm$3.09|
> |Carcinom|**93.77**$\pm$2.72|**93.77**$\pm$2.33|93.30$\pm$2.19|
> |Lung|**95.32**$\pm$1.02|94.09$\pm$1.80|95.24$\pm$0.61|
> |Lymphoma|**89.76**$\pm$2.20|**89.76**$\pm$2.44|**89.76**$\pm$1.52|
> |Over 11 datasets|**81.80**|81.33|81.64|
> ## Q5: Explain the modified SC in Eq.(4). Why $(d_i^{(l)}-\mathrm{u}^{(l)} )^3$ is used? How about other terms like $(d_i^{(l)}-\mathrm{u}^{(l)} )^1$.
> According to the literature (Linton, 2017), the **original formula of the SC** is $SC=\frac{\frac{1}{n}\sum_{i=1}^n(X_i-\bar{X})^3}{\left(\frac{1}{n}\sum_{i=1}^n(X_i-\bar{X})^2\right)^{3/2}}$, where $X_i$ are the individual data points, $\bar{X}$ is the mean and $n$ is the number of instances. As mentioned in lines 142-150 (right column), the modified SC can be obtained by substituting $\mathrm{u}^{(l)}$ and $\hat{\sigma}^{(l)}$ into the formula $SC$. That is, the term $(d_i^{(l)}-\mathrm{u}^{(l)})^3$ is directly derived from the term $(X_i-\bar{X})^3$ and has the same statistical meaning. The SC is a statistical measure that quantifies the asymmetry of a probability distribution. It indicates the degree to which data deviate from a symmetric, bell-shaped normal distribution **[1]**. Since we address the highest-density subclusters in the context of ADMHC, it is reasonable to use and modify the SC via Eq.(4).
>
> In addition, we appreciate the kind suggestion to discuss other terms such as $(d_i^{(l)}-\mathrm{u}^{(l)})^1$. We add the comparison between $(d_i^{(l)}-\mathrm{u}^{(l)})^3$ and $(d_i^{(l)}-\mathrm{u}^{(l)})^1$. To make the comparison reasonable, two terms with $(\cdot)^3$ in Eq.(4) are directly replaced by $(\cdot)^1$ (denoted as "w/ $(\cdot)^1$"). As demonstrated in **Rebuttal Table D**, our SIOFS using the term $(\cdot)^3$ achieves higher ACCs compared to $(\cdot)^1$. We will incorporate the above discussions into the final version.
> ## Q6: In Line 192, why normalize $\hat{s}^{(l)}$ by $\hat{s}^{(l)}/3$?
> Please see our response to **Reviewer aEFk, Q5**.

---

### Official Review · Reviewer_aEFk · 2025-03-14

**Overall Recommendation:** 3

**Summary:**

This paper proposes a supervised FS method, Stray Intrusive Outliers based FS (SIOFS), for data classification with intra-class ADMHC. By focusing on Stray Intrusive Outliers (SIOs), SIOFS modifies the skewness coefficient and fuses the threshold in the 3σ principle to identify the class body, scoring features based on the intrusion degree of SIOs.

Extensive experiments on 15 diverse benchmark datasets demonstrate the superiority of SIOFS over 12 state-of-the-art FS methods in terms of classification accuracy, normalized mutual information, and confusion matrix.

 SIOFS has the potential to improve analysis results in sectors that contain the classes with intra-class asymmetric instance distribution or multiple high-density clusters, from healthcare to finance. In all, SIOFS provides the theoretical foundation for further development of effective and interpretable models.

**Claims And Evidence:**

In Theorem 2, ‘the larger value in …’ is not rigorous, a better description is ‘if α1>α2∈（0，1）, then …’ It’s confusing to define 'larger'. Especially from experiments table 1, The α varies widely across datasets, ranging from 0.1 to 0.6. In the ablation experiment and the appendix, it is also shown that the change of α has a significant impact on the performance of some datasets. Is the final selection of a determined by the test results?

**Essential References Not Discussed:**

‘Feature selection techniques for machine learning: a survey of more than two decades of research.’ This article's descriptive classification of feature selection is very similar to the related work Review of FS in the appendix, but it is not cited. Of course this is not crucial, as the two most important and relevant articles are cited: FSDOC (Yuan et al., 2022) and IOFS (Yuan et al., 2024) methods.

**Experimental Designs Or Analyses:**

The elements of this experiment have been fully addressed; however, there are additional considerations regarding the deep learning component. While feature selection does not appear to offer significant advantages over deep feature extraction in terms of performance, as noted at the end of the article, its primary advantage over deep learning lies in interpretability—specifically, the ability to identify key factors influencing health outcomes rather than focusing solely on accuracy.

**Methods And Evaluation Criteria:**

The scheme designed in the experiment is meaningful and fair, and can explain the superiority of the algorithm to a certain extent. The designed evaluation methods are consistent with most related work, which reflects the rationality of the validation framework.

**Other Comments Or Suggestions:**

1. The discussion and expression need to be more rigorous
2. The experiment needs to be considered carefully

**Other Strengths And Weaknesses:**

The article demonstrates a high degree of originality and clarity in its narrative. However, the descriptions of theorems and key indicators are somewhat ambiguous, which poses challenges for code implementation. For instance, at line 161, "the most of" should be specified with a precise percentage (e.g., 0.8, 0.75, or 0.6). Additionally, the transition from 3σ to Formula 5, as well as the adjustment of the coefficient "2" mentioned in the left column at line 189, the normalization operation on line 193, all appear to be based on heuristic reasoning rather than rigorous derivation. Should such parameters be supported and fed back by experiments?

**Questions For Authors:**

I am particularly concerned about the selected value of α, specifically whether the experimentally determined optimal α aligns with the expectations derived from theoretical considerations. My primary concern is the potential discrepancy between theory and experimental results: the experimentally selected optimal α may not satisfy the criteria of "larger" and "the most of" as anticipated in the theoretical derivation. This concern has been exacerbated by some minor errors identified in this paper, which have raised doubts about the consistency between theory and practice.

Line 50 right column，class “2”, There is no class 2 in the figure 1 (b); line 71 left column: radii; line 445 left column: multipple.

**Relation To Broader Scientific Literature:**

From the recent review, the challenge of high-dimensional data classification has always existed, among which feature selection is a highly concerned scheme. In particular, asymmetric instance distribution and multi-density cluster, which are concerned in this paper, belong to strong prior knowledge, which can be used for model design with strong pertinently and predictable performance improvement. Recently, the use of outliers for feature selection [Yuan et al., 2022; Yuan et al., 2024] is a novel and niche research topic, which is beneficial to the development of machine learning. However, we hope to maintain the spirit of open source, so that more people can find the highlights of work and apply it to practice instead of doing closed development. Open-source code will not only increase your citations, it will also allow peers other than reviewers to review your work and provide more professional opinions.

**Theoretical Claims:**

Theorem 2 indicates that a larger α is necessary. However, the significant variation in α across different datasets presented in Table 1 raises concerns about the alignment between theoretical derivation and experimental validation. From the results in Figures 5 and 7, why does α differ so significantly for different data sets?

---

> ### Author Rebuttal · Authors · 2025-03-31
>
> We sincerely appreciate the reviewer’s feedback. Below, we address the concerns in detail.
> ## Q1: Some ambiguous descriptions about the theorems. Inconsistency between theory and experiment about α.
> As addressed in our response to **Reviewer 52hZ, Q1**, we clarify that the condition for in Theorem 2 does not contradict the experimental results. When $\alpha$ is small, the conclusion in lines 184-187 still holds. $\alpha$ has a significant impact on ACC because of the scattered distribution of clusters and multiple local high-density subclusters (see Fig. 1b). This problem can be solved by selecting more features.
>
> In addition, we regret the unclear descriptions. We delete "larger" and "the most of", and **revise Theorem 2** as follows: $d_1^{(l)},d_2^{(l)},\dots,d_{n_l}^{(l)}$, $\dots$. When $\alpha\in(0,1)$ and $\sigma^{(l)}>0$, $u^{(l)}+2\sigma^{(l)}>mode^{(l)}$ holds with probability 1 if $\hat{s}^{(l)}>0$, and $u^{(l)}+2\sigma^{(l)}<mode^{(l)}$ holds with probability 1 if $\hat{s}^{(l)}<0$. We will include the above details in the final version.
> ## Q2: Lack evidence or practical examples to validate the advantage on deep learning-based FS.
> We appreciate the constructive feedback provided. As mentioned in lines 417-418, deep FS has extensive applications, such as Remote Sensing (RS) scene classification and 3D object recognition. We add the ACC results by selecting all features (denoted as "AllFeat") in **Rebuttal Table B** and the CMs with the top 5% of features (see https://i.postimg.cc/HLw5K8z6/CMs-AID.png) on AID dataset. These results demonstrate that the SIOFS is superior to AllFeat while some comparative methods (such as ILFS, S^2DFS) is inferior. As shown in CMs, our SIOFS can further improve the classification performance on deep features against other methods. Combined with Fig.1a, for the "School" with intra-class ADMHC, the number of instances correctly classified by SIOFS is 28, while it is 24 by TRC and 27 by Fisher and ReliefF. SIOFS gains better performance in predicting the confusing classes, being able to identify key factors influencing RS monitoring. On UCM, SIOFS (94.95) also performs better than AllFeat (94.48); On ModelNet, SIOFS (93.03) has higher ACC than AllFeat (92.71). We will include these details in the final version.
>
> **Rebuttal Table B.** ACC (%) on AID. Six FS methods are randomly selected for comparison.
> ||ACC (%) $\uparrow$|
> |-|-|
> |*AllFeat*|*84.36*|
> |Fisher|84.92|
> |ReliefF|84.80|
> |TRC|85.05|
> |ILFS|83.93|
> |FSDOC|84.51|
> |S^2DFS|84.35|
> |SIOFS|**85.83**|
> ## Q3: Maintain the spirit of open source.
> We appreciate the valuable comment. The code will be released once accepted.
> ## Q4: Add an essential reference.
> As suggested, we will include this paper in the Related Work section in the final version.
> ## Q5: Explain the transition from 3σ to Eq.(5), the adjustment of the coefficient "2" in line 189, and the normalization in line 193.
> As mentioned in lines 129-131 (right column) and lines 179-187, and combined with the revised Theorem 2, the transition to Eq.(5) is a heuristic reasoning that follows the conclusion in lines 183-187 (see "That is, ..."), and it is necessary to introduce SC for normalization. The original formula of $SC$ (see our response to Reviewer 5k2J, Q5) and the literature [1] (see the response to Reviewer 5k2j, Q2) clarify that $s^{(l)}\in(-3,3)$ is an empirical guideline, providing intuitive thresholds to assess skewness severity and guide data adjustments, and reflecting realistic bounds for most practical datasets. That is, we have $\frac{1}{3}s^{(l)}\in(-1,1)$, then $2-\frac{1}{3}s^{(l)}\in(1,3)$. As mentioned in lines 198-205, the $2-\frac{1}{3}s^{(l)}$ with its range of (1,3) is more rational than 2 for $\sigma^{(l)}$ to address the highest-density subcluster in the context of ADMHC. As demonstrated in **Rebuttal Table C**, the proposed $2-\frac{1}{3}s^{(l)}$ outperforms other formulas in most cases, and the average ACCs over the 11 datasets further quantitatively prove the superiority of our method. We will incorporate the above discussions in the final version.
>
> **Rebuttal Table C.** Results of different adjustments and normalizations on some datasets. Our SIOFS is equipped with "$2-\frac{1}{3}s^{(l)}$". Additional settings are included in the table.
> |ACC (%) $\uparrow$|$2-\frac{1}{3}s^{(l)}$|$1-\frac{1}{3}s^{(l)}$|$3-\frac{1}{3}s^{(l)}$|$2-\frac{1}{2}s^{(l)}$|$2-\frac{1}{4}s^{(l)}$|
> |-|-|-|-|-|-|
> |CLL|**71.32**$\pm$3.84|69.52$\pm$1.42|66.37$\pm$1.24|70.57$\pm$2.53|69.52$\pm$4.58|
> |TOX|83.24$\pm$4.99|**83.63**$\pm$4.64|82.26$\pm$2.84|83.43$\pm$3.72|82.65$\pm$3.98|
> |Carcinom|**93.77**$\pm$2.72|93.10$\pm$1.05|92.82$\pm$2.14|92.72$\pm$2.36|93.10$\pm$1.96|
> |Lung|95.32$\pm$1.02|**95.63**$\pm$0.91|95.07$\pm$0.94|95.22$\pm$1.73|95.07$\pm$1.21|
> |Lymphoma|**89.76**$\pm$2.20|88.54$\pm$2.95|87.50$\pm$2.48|**89.76**$\pm$3.58|88.19$\pm$1.87|
> |Over 11 datasets|**81.80**|81.35|80.98|81.68|81.48|81.64|
> ## Q6: Minor errors.
> We will revise them in the final version.

---

> > ### Comment · Reviewer_aEFk · 2025-04-08
> >
> > Dear author, thank you for your reply. The imprecision of theory and experimental verification is still my concern. If there is empirical operation, is the theoretical derivation still rigorous? Because experiments can sometimes be deceptive.

---

> > > ### Author Response · Authors · 2025-04-09
> > >
> > > We sincerely appreciate the reviewer's feedback. We clarify that whether $\alpha$ is larger or smaller, it can be theoretically derived. The experimental results validate our theory. We summarize the details of derivations as follows.
> > >
> > > **Theoretical derivation about $\alpha$.** As mentioned in our submission, from the definition of RDM center (see Eq.3 and lines 137-139) and the fact that $\hat{s}^{(l)}$ has the same property of $s^{(l)}$ (see lines 154-157, right column), it is natural to deduce that (in lines 183-187) when $\alpha$ is smaller in (0,1), $\mathrm{u}^{(l)}+2\hat{\sigma}^{(l)}$ is relatively large for obtaining the highest density values in $d_1^{(l)},\dots,d_{n_l}^{(l)}$ when $\hat{s}^{(l)}>0$. Similarly, if $\hat{s}^{(l)}<0$, $\mathrm{u}^{(l)}+2\hat{\sigma}^{(l)}$ is relatively small for obtaining all highest density values in $d_1^{(l)},\dots,d_{n_l}^{(l)}$ (see **The detailed theoretical derivation when $\alpha$ is smaller**).
> > >
> > > To make this theory clear, we summarize as follows: For $d_1^{(l)},d_2^{(l)},\dots,d_{n_l}^{(l)}$, $\mathrm{u}^{(l)},\hat{s}^{(l)}$ are the same as in (4), and $mode^{(l)}$ is the same as the footnote of Section 3.1. When $\alpha\in(0,1)$ and $\hat{\sigma}^{(l)}>0$, $\mathrm{u}^{(l)}+2\hat{\sigma}^{(l)}>mode^{(l)}$ holds with probability 1 if $\hat{s}^{(l)}>0$, and $\mathrm{u}^{(l)}+2\hat{\sigma}^{(l)}<mode^{(l)}$ holds with probability 1 if $\hat{s}^{(l)}<0$.
> > >
> > > **The detailed theoretical derivation when $\alpha$ is smaller**: According to the definition of RDM center, $\mathrm{u}^{(l)}$ is the average of the top $\lceil\alpha\cdot n_l\rceil$ high density values in $d_1^{(l)},\dots,d_{n_l}^{(l)}$. When $\alpha$ is smaller in $(0,1)$ but $\lceil\alpha\cdot n_l\rceil=1$, we have $\mathrm{u}^{(l)}=mode^{(l)}$. Combined with $\hat{\sigma}^{(l)}>0$, we have $\mathrm{u}^{(l)}+2\hat{\sigma}^{(l)}>mode^{(l)}$. Following Theorem 1, let $\xi$ be the distance between an instance and the center of class $l$, $f(x)$ be the probability density function of $\xi$. Given the SC of $f(x)$ that $s^{(l)}>0$, there are two properties of statistical probability: (i) $f(mode+\varepsilon)>f(mode-\varepsilon)$, for any $\varepsilon>0$. (ii) $f(x)$ is monotonically increasing near the left side of $mode$ and decreasing near the right side. $d_1^{(l)},\dots,d_{n_l}^{(l)}$ is a random sample of $\xi$, let $d_2,d_3$ denote the 2nd and 3rd largest density in $d_1^{(l)},\dots,d_{n_l}^{(l)}$, respectively, and $d_1=mode^{(l)}$. Obviously there is $f(d_1)>f(d_2)>f(d_3)$. When $\alpha$ is smaller in $(0,1)$ but $\lceil\alpha\cdot n_l\rceil=2$, $\hat{\sigma}^{(l)}>0$ and $\hat{s}^{(l)}>0$. $\hat{s}^{(l)}$ has the same property of $s^{(l)}$ (see lines 154-157, right column). Due to random errors in the sampling, the probability that $d_2>d_1$ holds w.r.t property (i) is 1, then $\mathrm{u}^{(l)}=\frac{d_1+d_2}{2}>d_1$, i.e., at least $\mathrm{u}^{(l)}+2\hat{\sigma}^{(l)}>mode^{(l)}$ holds with probability 1. When $\alpha$ is smaller in $(0,1)$ but $\lceil\alpha\cdot n_l\rceil=3$, $\hat{\sigma}^{(l)}>0$ and $\hat{s}^{(l)}>0$. If $d_3>d_1$, then $\mathrm{u}^{(l)}=\frac{1}{3}(d_1+d_2+d_3)>d_1$; If $d_3<d_1$ (see https://i.postimg.cc/GmzqF1fV/PDF.png), since $f(d_3)<f(d_2)$, combining the property (i), $d_3$ can only be very close to $d_1$ in $\hat{s}^{(l)}>0$, namely $\mathrm{u}^{(l)}=\frac{1}{3}(d_1+d_2+d_3)$ is very close to $d_1$. Combined with $\hat{\sigma}^{(l)}>0$, it holds with probability 1 that $\mathrm{u}^{(l)}+2\hat{\sigma}^{(l)}>mode^{(l)}$ for $\lceil\alpha\cdot n_l\rceil=3$. When $\lceil\alpha\cdot n_l\rceil=4,5,\dots$, the same conclusion can be obtained. Similarly, if $\hat{s}^{(l)}<0$, $\mathrm{u}^{(l)}+2\hat{\sigma}^{(l)}$ is relatively small for obtaining all highest density values in $d_1^{(l)},\dots,d_{n_l}^{(l)}$.
> > >
> > > **Experimental verification for $\alpha$.** Based on the theory and explanations, the value of $\alpha$ is reasonable in $(0,1)$, and thus we set $\alpha=0.1,\dots,0.9$ in Fig. 7 and 9. As mentioned in lines 113-115 of our submission, $\alpha$ is the ratio of higher density to all instances in a class, it is affected by dataset characteristics. For some datasets, such as CLL (see Fig. 1b), due to the scattered distribution of clusters and multiple local high-density subclusters, $\alpha$ has a significant impact on ACC in some cases.
> > >
> > > The details of responses to reviewer's concerns are shown in the previous rebuttal. We will polish the complete mauscript, not limited to "larger" in Theorem 2 and "the most of" in line 178. We promise to release the code once it is accepted.
> > >
> > > As demonstrated by above explanations, our theoretical derivation is complete and rigorous (whether $\alpha$ is larger or smaller), and the experimental verification is effective. To avoid confusing, all above details will be included in the final version. We hope this is sufficient reason to consider raising the score.

---

### Official Review · Reviewer_52hZ · 2025-03-23

**Overall Recommendation:** 3

**Summary:**

For the problem of high-dimensional data classification with intra-class asymmetric instance distribution or multiple high-density clusters (ADMHC), a novel supervised feature selection (FS) method named Stray Intrusive Outliers-based FS (SIOFS) is proposed. The proposed method uses the RDM center to characterize the class body, and the modified skewness coefficient (SC) is adjusted and fused into the $3 \sigma$ principle to define the class body. Then, intrusion degree is modeled based on the conclusion of intersecting spheres. Finally, the feature ranking is determined by the intrusion degrees of SIOs. Experimental results demonstrate the effectiveness of the proposed method over other state-of-the-art FS methods.

**Claims And Evidence:**

$\bullet$ This paper mainly proposes the claim: In high-dimensional data classification with intra-class ADMHC, the distribution of distances is asymmetry or multi-peak. Existing FS methods rarely aim to identify the class body in the context of intra-class ADMHC. The proposed SIOFS method targets intra-class ADMHC for data classification.

$\bullet$ The theoretical results in this paper and the related experimental verification provide strong and clear evidence for the claim.

**Essential References Not Discussed:**

No. All the essential references have been adequately discussed.

**Experimental Designs Or Analyses:**

Yes. The experimental designs to verify the effectiveness of the proposed method are complete, and the comparison of other state-of-the-art FS methods and the comparison and analysis of experimental results on several diverse benchmark datasets well demonstrate the effectiveness of SIOFS.

**Methods And Evaluation Criteria:**

Yes. The proposed method indeed demonstrates advantages over other state-of-the-art FS methods in experiments.

**Other Comments Or Suggestions:**

Line 161, "Condition i" --> "Condition (i)",

Line 123, right column, "condition ii" --> "condition (ii)",

Line 130, right column, "condition ii" --> "condition (ii)".

**Other Strengths And Weaknesses:**

$\bullet$ Strengths:

The proposed method effectively handles the data classification problem with intra-class ADMHC and achieves better performance than existing methods.

$\bullet$ Weaknesses:

1. Theorem 2 assumes that $\alpha$ is a larger value, but experimental results show that the value of $\alpha$ is often relatively small, with the largest being $0.6$ that occurs by chance. What is the exact range of values ​​of $\alpha$ in Theorem 2? How to explain the inconsistency between theoretical results and experiments?

2. In Equation (6), the coefficient $2$ is adjusted to $2-\frac{1}{3} \hat{s}^{(l)}$, and the motivation and basis for this adjustment are not explained in detail. Can coefficient $\frac{1}{3}$ of $\hat{s}^{(l)}$ be adjusted to other values ​​between 0 and 1?

**Questions For Authors:**

For classification problems, the increase in the number of classes will have a significant impact on the performance of the method, that is, it will increase the difficulty of classification. I noticed that the number of classes in the datasets involved in the experiments is actually relatively small, with the largest number of only $40$ classes. A natural confusion is whether the proposed method can still have a significant advantage over existing methods when the number of classes is large? Therefore, as the number of classes increases, the effectiveness of the proposed method needs further experimental verification.

**Relation To Broader Scientific Literature:**

Prior FS methods rarely aim to identify the class body in the context of intra-class ADMHC. This paper explains the motivation for quantifying the intrusion degree of SIOs and proposes the SIOFS method to deal with intra-class ADMHC for data classification.

**Theoretical Claims:**

Yes. I have reviewed the theoretical proofs and the correctness of the proofs supports the claim.

---

> ### Author Rebuttal · Authors · 2025-03-30
>
> We sincerely thank Reviewer 52hZ for the recognition of our work and for providing constructive comments.
>  ## Q1: Explain the inconsistency between Theorem 2 and results about $\alpha$.
> Sorry for the incomplete statement about Theorem 2. We clarify that the condition for $\alpha$ in Theorem 2 does not contradict the experimental results. **When $\alpha$ is small, the conclusion in lines 183-187 (left column) still holds**. For easy understanding, we add the following statement.
>
> According to the definition of RDM center (see lines 96-98, right column), $\mathbf{u}^{(l)}$ is the average of the top $\lceil\alpha\cdot n_l\rceil$ high density values in $d_1^{(l)},\dots,d_{n_l}^{(l)}$. When $\alpha$ is smaller but $\lceil \alpha\cdot n_l\rceil=1$, we have u$^{(l)}=mode^{(l)}$. Following Theorem 1, let $\xi$ be the distance between the instance and the center of class $l$ and f(x) be the probability density function of $\xi$. Assume that the SC of $f(x)$ is greater than 0. There are two properties of statistical probability (Harris & Stocker, 1998): (i) $f(mode+\varepsilon)>f(mode-\varepsilon)$, for any $\varepsilon>0$. (ii) f(x) is monotonically increasing near the left side of the $mode$ and decreasing near the right side. In Theorem 2, $d_1^{(l)},\dots,d_{n_l}^{(l)}$ is a random sample of $\xi$, let $d_2,d_3$ denote the second and third largest density in $d_1^{(l)},\dots,d_{n_l}^{(l)}$, respectively, and $d_1=mode^{(l)}$. Obviously there is $f(d_1)>f(d_2)>f(d_3)$. When $\alpha$ is small but $\lceil\alpha\cdot n_l\rceil=2$, $\hat{\sigma}^{(l)}>0$ and $\hat{s}^{(l)}>0$. Given the chance of errors in the sampling process, the probability that $d_2>d_1$ holds according to condition (i) is 1, then u$^{(l)}=\frac{d_1+d_2}{2}>d_1$, i.e., at least u$^{(l)}+2\hat{\sigma}^{(l)}>mode^{(l)}$ holds with probability 1. When $\alpha$ is small but $\lceil\alpha\cdot n_l\rceil=3$, $\hat{\sigma}^{(l)}>0$ and $\hat{s}^{(l)}>0$. If $d_3>d_1$, then u$^{(l)}=\frac{1}{3}(d_1+d_2+d_3)>d_1$; If $d_3<d_1$ (see figure https://i.postimg.cc/GmzqF1fV/PDF.png), since $f(d_3)<f(d_2)$, combining the property (i), $d_3$ can only be very close to $d_1$ in this skewed distribution, i.e., u$^{(l)}=\frac{1}{3}(d_1+d_2+d_3)$ is very close to $d_1$. Combined with $\hat{\sigma}^{(l)}>0$, it holds with probability 1 that u$^{(l)}+2\hat{\sigma}^{(l)}>mode^{(l)}$ for $\lceil \alpha \cdot n_l\rceil=3$. When $\lceil\alpha\cdot n_l\rceil=4,5,\dots$, the same conclusion can be obtained. Similarly, when $\hat{s}^{(l)}<0$ and $\alpha$ is small, but $\lceil\alpha\cdot n_l\rceil=1,2,\dots$, u$^{(l)}+2\hat{\sigma}^{(l)}<mode^{(l)}$ holds with probability 1.
>
> In addition, we rewrite "Based on Theorem 2..." (line 181) as "Combining Theorem 2..." to make the discussion and expression more rigorous. Based on the above explanations and Theorem 2, the value of $\alpha$ is reasonable in (0,1), and thus we set $\alpha=0.1,\dots,0.9$ in Figures 5 and 7. Since $\alpha$ is the ratio of higher density to all instances in a class (see lines 113-115), it is affected by different dataset characteristics. For example, due to the scattered distribution of clusters and multiple local high-density subclusters in CLL (see Fig. 1b), $\alpha$ has a significant impact on ACC (see Fig. 7). This problem can be solved by selecting more features. Above statements will be added in the final version.
> ## Q2: Explanations of the adjustment of the coefficient "2" mentioned in the left column at line 189 and the normalization operation on line 193.
> Please see our response to **Reviewer aEFk, Q5**.
> ## Q3: Typos: conditions i-->(i), ii-->(ii) and iii-->(iii).
> As suggested, we go over the paper and will revise them in the final version.
> ## Q4: Whether SIOFS can still have a significant advantage over other methods when the number of classes is large?
> As suggested, we add comparisons with the baselines on Caltech101 dataset, which has 101 classes and 3030 images. Like (Yuan et al., 2022), we also use the Fisher Vectors (262144 dimensions). Additionally, in order to reduce the computation time and preserve the main properties of the original representation, we uniformly sample these very high dimensional feature vectors with 50 components, i.e. $1,51,\dots,262144$, and obtain the final feature vector (5243 dimensions) for each image. Following the setting in (Yuan et al., 2022), we select 60%,70%,80%,90% of features. Same with the original paper, ACC and NMI are calculated over all baselines. And SIOFS outperforms all comparative methods in all cases. Some results are shown in **Rebuttal Table A**.
>
> **Rebuttal Table A.** ACC (%) on the large-scale high-dimensional Caltech101 dataset. Details will be added in the final version.
> |ACC (%) $\uparrow$|60%|70%|80%|90%|
> |-|-|-|-|-|
> |FSDOC|43.56|43.43|43.93|43.20|
> |S2DFS|42.38|43.43|43.56|43.66|
> |IOFS|45.45|48.84|51.52 |53.83|
> |EGCFS|43.96|43.89|43.86|44.26|
> |FSDK|39.14|40.46|42.15|42.97|
> |SIOFS|**45.74**|**48.98**|**51.58**|**54.13**|

---

### Decision · Program_Chairs · 2025-05-01

**Decision:**

Accept (poster)

**Comment:**

This paper proposes a novel Stray Intrusive Outliers-based Feature Selection (SIOFS) method for high-dimensional data classification with intra-class asymmetric instance distribution or multiple high-density clusters (ADMHC). The key innovation lies in focusing on stray intrusive outliers that intrude into other class bodies, using a refined density-mean center to characterize class bodies and modifying the skewness coefficient fused with the 3σ principle to identify class boundaries. Theoretical guarantees are provided for the proposed method, and extensive experiments on 15 benchmark datasets demonstrate its effectiveness.

The paper received generally positive reviews, with all three reviewers acknowledging its ​​novelty​​ in addressing ADMHC scenarios and its ​​strong empirical results​​. Reviewer aEFk highlighted the method's potential applications but initially expressed concerns about theoretical-experimental alignment. While minor weaknesses were noted, the reviewers unanimously leaned toward acceptance after the rebuttal.